# Quantifying Contribution of Atmospheric Circulation to Precipitation Variability and Changes in the U.S. Great Plains and Southwest Using Self Organizing Map – Analogue

Yizhou Zhuang[1], Rong Fu[1]

[1]Department of Atmospheric and Oceanic Sciences, University of California, Los Angeles, CA 90095, USA

*Correspondence to*: Yizhou Zhuang (zhuangyz@atmos.ucla.edu)

**Abstract.** The Great Plains and Southwest regions of the U.S. are highly vulnerable to precipitation-related climate disasters such as droughts and floods. In this study, we propose a self-organizing map–analogue (SOMA) approach to empirically quantify the contribution of atmospheric moist circulation (mid-tropospheric geopotential and column moisture transport) to the regional precipitation anomalies, variability, and multi-decadal changes. Our results indicate that moist circulation contributes significantly to short-term precipitation variability, accounting for 54-61% of the total variance in these regions, though these contributions vary significantly across seasons. As indicated in previous research, Pacific Decadal Oscillation (PDO) is one of the major climate modes influencing the long-term multi-decadal variation of precipitation. By contrasting three multi-decadal periods (1950-1976, 1977-1998, 1999-2021) with shifting PDO phases and linking the phase shift to SOM nodes, we found that circulation changes contribute considerably to the multi-decadal changes of precipitation anomaly in terms of the mean and days of dry and wet extremes, especially for the Southern GP and Southwest. However, these circulation-induced changes are not totally related to the PDO phase shift (mostly less than half) since internal variability or anthropogenically induced changes in circulation can also be potential contributors. Our approach improves upon flow analogue and SOM-based methods and provides insights into the contribution of atmospheric circulation to regional precipitation anomalies and variability.

## 1 Introduction

The United States Great Plains (GP) and Southwest (SW) are central to national and global agricultural production (Elias et al., 2016; Parton et al., 2015). As the drier half of the continental United States (CONUS) with high variability of precipitation, the GP and SW rely on irrigation much more heavily than the eastern states, and they are known to be susceptible to climate disasters such as extreme droughts, wildfires, storms, and flooding, which are all listed among the top billion-dollar weather and climate disasters (NCEI, 2023). Understanding the driving factors and mechanisms for the occurrence of precipitation anomalies is crucial for the evaluation and improvement of current prediction systems as well as water resource management.

Past studies have reported various mechanisms that link to precipitation variability in North America, including anthropogenic forcings (Kirchmeier-Young and Zhang, 2020), internal atmospheric variabilities (Mckinnon and Deser, 2021), El Nino/Southern Oscillation (ENSO) (Hu and Huang, 2009), Pacific Decadal Oscillation (PDO)

(Lehner et al., 2018), Pacific-North America (PNA) (Zhuang et al., 2021a), Atlantic sea surface temperature (Hu and Feng, 2008), each of which can contribute differently for different locations, time scales, seasons, or even individual events. These internal variabilities and large-scale remote forcings can influence the local precipitation either by affecting the atmospheric circulation conditions above or through local and remote land surface feedbacks, both of which control the vertical air motion and moisture supply that finally lead to precipitation.

Many recent studies have argued that the long-term precipitation changes in these regions, at least from the mid-20th century until now, are primarily driven by PDO-induced atmospheric teleconnection to North America, instead of anthropogenic warming, as evidenced by the relatively dry periods with negative PDO phase (1940s-1970s and 1999 onwards) when many of the major drought events occurred (Hoerling et al., 2014; Seager et al., 2014; Mankin et al., 2021), and a relatively wet period with positive phase (1980s-1990s). Yet, it is not clear quantitatively how much these long-term changes in precipitation can be explained by the PDO-related circulation variability.

The self-organizing map (SOM) or Kohonen Map (Kohonen, 1990), an unsupervised neural network-based clustering method, has long been demonstrated as an effective and powerful tool to study dynamics or atmospheric circulation patterns related to precipitation variability, drought, or other atmospheric/oceanic phenomena (Liu and Weisberg, 2011; Zhuang et al., 2020). Different from other clustering methods, such as K-means and hierarchical clustering, SOM has the advantage of sorting samples into types (SOM nodes) connected to each other in a "map" that preserves the topological structure of the data, so similar types are close to each other. Based on SOM, Cassano et al. (2007) proposed a quantitative partitioning method (referred to as "C2007" hereafter) to determine the circulation contributions to anomalies or trends of surface parameters such as precipitation. The idea is that the total precipitation anomaly for a certain node $K$ during a period ($P'_K$) can be decomposed as the sum of a circulation or dynamic component ($P'_{K,dyn}=\Delta f_K P'_K$) controlled by the mean frequency change of SOM nodes, a thermodynamic component ($P'_{K,the}=f_K \Delta P'_K$) related to the mean state change of precipitation for the same SOM node, and an interaction term ($P'_{K,int}= \Delta f_K \Delta P'_K$) related to both. Detailed descriptions of this method can be referred to Cassano et al. (2007) and Horton et al. (2015). However, this partitioning method assumes that $P'_{dyn}$ at the time scale of individual sample (e.g., daily) stays the same for circulation patterns classified to the same SOM node. This underlying assumption leads to the result that the determination of $P'_{dyn}$ could be sensitive to the choice of SOM node number. An additional inconvenience related to this method is that the SOM analysis is often conducted in certain seasons as the circulation–precipitation relationship could vary in different seasons.

Flow analogue or dynamic adjustment (e.g., Deser et al., 2016; Horowitz et al., 2022; Jezequel et al., 2018; Lehner et al., 2018; Terray, 2021; Yiou et al., 2007) is another widely used method (referred to as analogue hereafter) to determine the circulation contribution. For a certain day (or other time scales), different from the C2007 method, the constructed flow analogue (Lehner et al., 2018) method searches for a limited number of days (analogue days) with circulation patterns most similar to that of the target day, instead of days assigned to the same SOM node in C2007; it then combines precipitation anomaly of these analogue days to form a final analogue anomaly or dynamic component ($P'_{dyn}$), and the residual term which cannot be explained by circulation is defined as the thermodynamic component ($P'_{the}$). Detailed steps and variants of the analogue method can be found in the literature (e.g., Zhuang et

al., 2021b). The analogue theoretically better estimate the dynamic contribution by utilizing circulation patterns from the analogue days and accounting for their subtle difference.

In this study, we proposed a modified approach to quantify the multivariate circulation contribution to $P'$ in terms of its daily variability, which combines the advantages of both SOM and constructed analogue methods. We further use this approach to quantify the contribution of overall circulation changes and potentially PDO-related circulation changes to the long-term multidecadal changes of $P'$.

The datasets and methods used in this study are described in section 2. All the main results, including the SOM inferred relationship between large-scale atmospheric circulation and precipitation and SOM – analogue based quantification of circulation contributions to daily variability and long-term changes of precipitation anomalies, are presented in section 3. Section 4 summarizes the main conclusions of this study and also gives a brief discussion about the implications of this study and limitations of our methodology and results.

## 2 Data and Method

### 2.1 Data

In previous studies related to SOM and analogue, large-scale circulation is generally represented by mean sea level pressure (SLP) or geopotential height at 500 hPa (Z500). Here, we choose Z500 over SLP as our experiments have demonstrated analogues derived from Z500 show greater similarity in synoptic variability with observed surface anomalies and yield smaller residuals compared to analogues derived from SLP (Zhuang et al., 2021b). Additionally, impacts of anomalous atmospheric moisture transport represented by vertically integrated vapor transport (IVT) on precipitation variability and drought are also suggested in the literature (Zhuang et al., 2020; Pu et al., 2016; Swales et al., 2016). The 3-hourly Z500 and IVT data at 1°×1° resolution are obtained from the fifth generation ECMWF atmospheric reanalysis of the global climate (ERA5) (Hersbach et al., 2020). We use daily averages for all months from 1950 to 2021 over the CONUS area (130°-70°W, 25°-55°N) as input for later analyses. To account for the thermal expansion of the warming atmosphere, we subtract the daily global area-weighted mean Z500 from the daily Z500 data at each grid point so that there is no linear trend in the Z500 data due to warming (Christidis and Stott, 2015; Siler et al., 2019; Zhuang et al., 2021b); however, this procedure does not remove any anthropogenically forced changes in atmospheric circulation patterns. Other studies (e.g., Sippel et al. 2019) also use linear detrending or high-pass filtering to preprocess Z500 data at each grid point, assuming that the forced circulation response was smooth and additive; we do not employ these approaches due to our short observational record and they could remove some decadal circulation trend due to internal variability (Zhuang et al. 2021b). To mitigate high-frequency synoptic noise, we employ a simple 5-day moving average filter to both Z500 and IVT. Subsequently, we calculate the daily standardized anomaly (Z500′ and IVT′) using the 5-day moving average-filtered data relative to the 1950-1999 climatology. We select the 1950-1999 period as the climatology reference to maximize the utilization of available data for this study; furthermore, this period, which predates the significant warming trend, typically serves as a robust baseline with less climate change impact. In addition, we adopt the running mean approach used in Zhuang et al. (2020) to calculate a smoothed daily climatological mean and standard deviation.

For precipitation, the Climate Prediction Center (CPC) global unified gauge-based analysis of daily precipitation ($P$) is used. This dataset is at 0.25°×0.25° resolution and covers the period from 1948 to the present. In this study, we mainly focus on three regions, including the Southern GP (SGP; 105°-92°W, 26°-38°N), the Northern GP (NGP; 105°-92°W, 38°-49°N), and the Southwestern US (SW; 114°-105°W, 31°-42°N). Therefore, area-weighted mean precipitation is calculated over these three regions. A 5-day moving average filter is also applied to precipitation similar to the circulation data. The precipitation anomaly ($P'$) is calculated relative to 1950-1999 climatology using the same approach stated above for circulation, but without the standardization process as precipitation generally follows a gamma distribution rather than a Gaussian distribution.

Besides circulation and precipitation, other data used in this study including ERA5 2-m dewpoint temperature (D2), 700hPa temperature (T700) data, and convective available potential energy (CAPE), as well as the monthly Pacific Decadal Oscillation (PDO) index provided by the National Center for Environmental Information (NCEI). The D2 and T700 data are used to calculate the convective inhibition (CIN) index (CINi = D2 – T700) (Myoung and Nielsen-Gammon, 2010) to represent the lower atmospheric instability; ERA5 also provides CIN data but it is not always available around the year due to its original definition from the parcel buoyancy model. Both CAPE and CINi data are processed the same way as circulation data to acquire their standardized anomalies (CAPE′ and CINi′) for further analyses. The PDO index is one of the most important Pacific climate variability and is often described as an El Nino-like pattern but with long-term persistence up to several decades (Zhang et al., 1997). The NCEI monthly PDO index is calculated based on the NOAA extended reconstruction of sea surface temperature version 5 (ERSSTv5). 3-month running mean PDO index is used in this study to smooth out some monthly SST fluctuations and reduce short-term noise. Our whole analysis period 1950-2021 can be divided into three periods, each with a different PDO phase from its previous one, including P1: 1950-1976 (negative), P2: 1977-1998 (positive), and P3: 1999-2021 (negative). Long-term change of precipitation in terms of its mean or days of extremes for our following analysis (section 3.3) is defined as the difference between two adjacent periods (P2 – P1 and P3 – P2).

## 2.2 Method

### 2.2.1 Multivariate SOM

In this study, large-scale atmospheric circulation condition is jointly represented by Z500′ and IVT′, as both of them have been demonstrated to be important for the precipitation variability over North America. The data matrices of these two variables are concatenated along the spatial dimension, so the input data matrix has a size of ($2 \times N_{lon} \times N_{lat}$, $N_t$). Specifically for this study, $N_{lon} = 61$, $N_{lat} = 31$, and $N_t = 365$ days/year × 72 years (leap days removed for simplicity), so the input size is (3782, 26280).

To speed up the SOM training process, we employ a principal component analysis (PCA) preprocessing technique (Zhuang et al., 2020). PCA or Empirical Orthogonal Functions (EOF) is a widely used approach for data dimension reduction or extracting spatial modes of variability (Reusch et al., 2005). Here, PCA is used to decompose the high dimensional circulation data (two variables and thousands of grid points) into the matrix multiplication product of some low dimensional principal components (PCs) and their corresponding loading vectors (modes). By only retaining the top PCs and modes that explain the majority of the variance in the original data, we can use these PCs as reduced-

size input data for SOM. Specifically in this study, we retain the top 26 PCs which explain 87.3% of the combined variance in Z500′ and IVT′ (or 90.6% and 77.4% of the variance in Z500′ and IVT′, respectively).

Determination of SOM node number and evaluation of the trained SOM quality are often complicated issues. Too few SOM nodes could lead to an underrepresentation of samples, while too many nodes likely result in redundancy and create difficulties for analysis and visual interpretation. In this study, we have tested multiple SOM schemes with various node numbers up to 200. Three metrics are calculated for the trained SOM of each scheme to assess the SOM quality, including the quantization error (QE), topographic error (TE), and combined error (CE). QE represents the average Euclidean distance between each data vector and the SOM node it is assigned to, or the so-called best matching unit (BMU). TE is the proportion of all input data vectors with first and second BMUs not adjacent to each other in the SOM map. CE is a metric that combines the concepts of both QE and TE and was defined by Kaski & Lagus (1996).

SOM training and error metric calculation are performed using the Matlab SOM Toolbox (Vatanen et al., 2015). Training length is fixed at 100 epochs (50 for rough training and 50 for fine-tuning) for all schemes. Additional experiments show that increasing training length can reduce QE, but the improvement is very marginal after around 100 epochs (not shown). In the SOM Toolbox, after inputting a desired node number, the SOM map shape is automatically determined. During the initialization before the training, we employ the default setting, i.e., linear initialization, where weight vectors are initialized in a linear manner along the subspace defined by the two principal eigenvectors of the input dataset. This choice is made to facilitate the reproducibility of our results. While SOM is generally robust to initialization, slight variations in outcomes may occur when using random initialization, where the weight vectors are initialized with random small values. To assess the impact of initialization on SOM error metrics, we also conduct 100 additional SOM training with random initialization for each node number setting and analyze the range of their error metrics. After training SOMs for all schemes with various node numbers and calculating their quality metrics, the result in Fig. 1 shows that overall, QE decreases with node number as more nodes mean more detailed classification of circulation patterns and thus are more representative of individual samples; TE increases with node number as a more complex network often leads to a larger topographic structure; CE, which combined the concepts of both quantization and topographic errors, decreases with node number first when the number is relatively small and then exhibits larger fluctuation than QE and TE. A 7×4 SOM scheme is chosen for our following analysis as this scheme has a relatively smaller node number that facilitates the visualization and analysis of the results and also has a lower CE compared to schemes with similar node numbers.

### 2.2.2 Empirical quantification of moist circulation contribution: SOM-Analogue (SOMA) Method

As stated in the introduction section, the C2007 dynamic/thermodynamic partition method assumes the mean $P'$ of all days with the same BMU node as its dynamic component $P'_{dyn}$. However, this approach can underestimate variability in true $P'_{dyn}$, especially when the node number is small and the circulation pattern of a BMU is less representative of the assigned samples. On the other hand, the flow analogue method (e.g., Yiou et al. 2007; Zhuang et al. 2021b), estimates $P'_{dyn}$ for a given day by identifying historical days with similar atmospheric circulation conditions (analogue days) and combining the $P'$ information from these selected analogue days. Specifically, the flow analogue method

employed by Zhuang et al. (2021) follows three main steps. First, for a particular day, analogue days with similar circulation conditions are chosen by minimizing a distance function between the circulation field (e.g., Z500′) on the target day and that on historical days within a specific calendar range centered around the target day. Second, the top $N$ analogue patterns with the smallest distance function are linearly combined to create a "constructed analogue" pattern that resembles the circulation pattern of the target day. Finally, the $N$ coefficients from the linear combination are applied to $P'$ values of the $N$ analogue days to obtain $P'_{\text{dyn}}$. The constructed flow analogue method generally explains a larger portion of the observed $P'$ variance and its estimate is not sensitive to SOM node number, so it has a higher accuracy in capturing the complex relationship between atmospheric circulation and precipitation anomalies and theoretically provides a better estimate of circulation contribution compared to the C2007 method. This improvement is primarily due to the flow analogue method considering the differences among circulation patterns in individual analogue days and combining $P'$ values from these analogue days. However, flow analogue is less effective than the C2007 method when evaluating the relationship between a specific circulation type and $P'$.

In this study, we propose a new method called SOM-Analogue (SOMA) which leverages the strengths of both SOM (C2007) and flow analogue. SOMA aims to be capable of quantifying overall circulation contribution and the contribution of a specific circulation type to $P'$, and also provide a more robust circulation contribution estimate that is less sensitive to parameter choices and explains a large portion of observed $P'$ variability. It's worth noting that attempting to separate the actual dynamic and thermodynamic components solely through a circulation clustering approach like SOM can be challenging. Each type of circulation, as represented by SOM nodes, inherently encompasses thermodynamic responses. Therefore, the distinction between "dynamic" and "thermodynamic" components can be ambiguous when using these terms. Consequently, we prefer to refer to the precipitation anomaly influenced by moist circulation patterns involving Z500 and IVT as $P'_{\text{SOMA}}$ instead of $P'_{\text{dyn}}$, and the residual part that cannot be explained by circulation as $P'_{\text{RES}}$ instead of $P'_{\text{the}}$. This emphasizes that our results regarding circulation contributions are contingent on our chosen set of circulation variables.

The basic idea behind SOMA is to incorporate variability in $P'_{\text{SOMA}}$ for days with the same BMU node, similar to the flow analogue method, while adhering to the C2007 framework. In contrast to the flow analogue method, where analogue days are determined by minimizing the Euclidean distance of anomalous circulation patterns, in SOMA, the analogue days are directly selected from the daily samples sharing the same BMU. Additionally, the constructed flow analogue approach relies on a linear combination of precipitation anomaly from analogue days to quantify circulation contribution at a target day $j$, i.e., $P'_{j,\text{SOMA}} = \sum_i a_i P'_i = \sum_i a_i P'_{i,\text{SOMA}} + \sum_i a_i P'_{i,\text{RES}}$, where the coefficients $a_i$ are determined by the linear dependency of the circulation pattern on target day $j$ and those on analogue days $i$, i.e., $Z'_j = \sum_i a_i Z'_i$. However, it is important to note that the analogue day precipitation anomaly $P'_i = P'_{i,\text{SOMA}} + P'_{i,\text{RES}}$ includes both a circulation-contributed component and a residual component. This approach overlooks the potential thermodynamic influences in the precipitation anomaly on analogue days, and the resulting circulation-contributed component $P'_{j,\text{SOMA}}$ still retains a thermodynamic residual $\sum_i a_i P'_{i,\text{RES}}$, which may not be negligible with limited samples. To mitigate this issue, we make the assumption that for each group of similar circulation pattern (same SOM node), $P'_{j,\text{SOMA}} = \sum_i a_i P'_{i,\text{SOMA}}$. This differs from the equation form in constructed flow analogue and excludes the term $\sum_i a_i P'_{i,res}$. Following constructed flow analogue, where each circulation pattern is linearly dependent on other

circulation patterns within the same SOM node, i.e., $Z'_j = \sum_i a_i Z'_i$, we can see that $P'_{\text{SOMA}}$ and $Z'$ share the same linear form and coefficients. Therefore, $P'_{\text{SOMA}}$ and $Z'$ can be considered linearly dependent as well, allowing us to treat the calculation of $P'_{\text{SOMA}}$ as a regression problem, which is different from the circulation analogue in previous studies (e.g., Deser et al. 2016; Lehner et al. 2018). Detailed steps are outlined below.

1) After PCA preprocessing, the original circulation anomaly matrix $\mathbf{Y}_{Np \times Nt}$ can be written as $\mathbf{Y}_{Np \times Nt} = \mathbf{V}_{Np \times Ne} \cdot \mathbf{X}_{Nt \times Ne}^{\text{T}}$, where $N_p$ is the number of grid points (61×31) multiplied by the number of circulation variables (2), $N_t$ the number of days, and $N_e$ the number of all PCs. By only retaining the top $K$ PCs, $\mathbf{Y}_{Np \times Nt} \approx \mathbf{V}_{Np \times K} \cdot \mathbf{X}_{Nt \times K}^{\text{T}}$.

2) The PC matrix $\mathbf{X}_{Nt \times K}$ is used as input to train a SOM. For each daily sample $\mathbf{x}_t$ (1×$K$) at time $t$, we can find its corresponding BMU, i.e., $\text{BMU}(\mathbf{x}_t) = \mathbf{B}_n$, where $\mathbf{B}_n$ is one of the SOM nodes, and $n$ ($n = 1, 2, 3, …, N$) can be determined by minimizing the Euclidean distance between $\mathbf{x}_t$ and $\mathbf{B}_n$. The circulation pattern for node $n$ can be then recovered as $\mathbf{V}_{Np \times K} \cdot \mathbf{B}_n^{\text{T}}$.

3) For any day $t_0$, assuming its BMU is node $n$, we find all days $t_i$ ($i = 1, 2, …, M$) that have this node $n$ as their BMU and are within the 91-day calendar window centered on the day $t_0$ but not in the same year as $t_0$, to account for the seasonal dependence of circulation-precipitation relationship, i.e., $\text{BMU}(\mathbf{x}_{ti})=\text{BMU}(\mathbf{x}_{t0})$, $|\text{DOY}(t_i) – \text{DOY}(t_0)| \leq 45$ d, $\text{Year}(t_i) \neq \text{Year}(t_0)$.

4) Solve the regression problem: $\mathbf{P}_{1 \times M} = \mathbf{U}_{1 \times K} \cdot \mathbf{X}_{M \times K}^{\text{T}}$, where $\mathbf{P}_{1 \times M}$ is the precipitation vectors for days $t_i$ ($i = 1, 2, …, M$) from the previous step, $\mathbf{X}_{M \times K}$ is input PC samples for these $M$ days. The regression coefficient vector $\mathbf{U}_{1 \times K}$ is then used to calculate the dynamic precipitation anomaly for the day $t_0$, $P'_{\text{SOMA},t0} = \mathbf{U}_{1 \times K} \cdot \mathbf{x}_{t0}^{\text{T}}$.

5) Repeat steps 3) and 4) for all days until all $P'_{\text{SOMA}}$ are calculated.

To account for the sampling uncertainty, we apply a bootstrap technique in step 3) where we create multiple data sets of $\mathbf{X}_{M \times K}$ by randomly sampling $\mathbf{x}_{ti}$ with replacement. After bootstrapping and repeating steps 3) to 5) for 1000 times, the 1000 calculated $P'_{\text{SOMA},t0}$ are then averaged as the final result.

## 3 Result

### 3.1 Atmospheric circulation pattern clustering and link to precipitation

After the 7×4 SOM for Z500′ and IVT′ is trained and each daily sample is assigned to its BMU node, circulation patterns with the same BMU node are averaged as the general representation of circulation condition for the particular node. The composite circulation anomaly patterns for all SOM nodes are shown in Fig. 2. Similar to our earlier study (Zhuang et al., 2020), the SOM map shows a continuum of anomalous circulation and moisture transport patterns over the CONUS. Regions with large Z500′ gradients often correspond to large IVT′ due to the geostrophic balance relationship. For example, when an anomalous geopotential low is centered over the western or central US and a geopotential high over the eastern or southeastern US (e.g., nodes A1, B1, C1, C2, D1, D2, and D3), enhanced moisture transport from the Gulf of Mexico represented by the large positive IVT′ is often observed in-between the geopotential low and high with elevated geopotential gradient.

As the SOM is trained with daily samples of all seasons and considering the potential seasonal variation of occurrence frequency and circulation-precipitation relationship, we calculate the frequency and average $P'$ for each SOM node

during every 91-day calendar window during all years. The resulting seasonal variations of node frequency and mean $P'$ are shown in Fig. 3 and Fig. 4, respectively. In Fig. 3, some nodes exhibit very clear seasonal variation with the range of frequency being as large as 4% (mean frequency and seasonal frequency range for each node is also available in Fig. 2); they either peak in cold seasons, such as C1 and D1, with the anomalous geopotential high centered more to the southeastern US and strong moisture transport to the SGP, or peak in the warm seasons, such as C4 and D4 where the high is centered more to the northeastern US so the moisture transport is weaker and more towards the north. Some other nodes show relatively stable frequency throughout the year, with a range of less than 1%, such as nodes C6, C7, and D7 where the CONUS is mostly dominated by an extensive geopotential high.

Similarly for precipitation, it also shows apparent seasonal variations for certain nodes (Fig. 4). For example, node D1, located at the lower-left corner of the SOM, brings the strongest positive IVT′ from the Gulf of Mexico and the Gulf of California of all nodes, and it is one of the major wet nodes related to positive $P'$ in all three regions during the cold season. However, during the warm season, node D1 is more of a neutral or weakly dry node as it is linked to a small negative $P'$ in the SGP; for the NGP, this node is still a wet node and the average $P'$ is larger than that during the cold season. This can in part be explained by the seasonal difference in precipitation mechanisms. In the warm season, precipitation is mostly convective-driven, reducing the impact of large-scale vertical air motion associated with certain circulation patterns. Additionally, the composite CAPE′ and CINi′ map for node D1 (Fig. S1) shows relatively smaller CAPE′ and a more stable lower troposphere (negative CINi′) compared to other nodes in the warm season (May to July), suppressing convective development thus limiting precipitation regardless of the strong moisture transport, whereas in the cold season (November to January), CAPE′ and CINi′ shows larger positive values associated with more precipitation compared to other nodes. However, it's important to note that the above comparison is within the same season, as node D1 in the warm season still corresponds to higher traditional CAPE and CINi values (not anomalies), as well as less total precipitation, compared to the cold season. Similar situations occur to many other nodes, either exhibiting opposite signs of mean $P'$ in different seasons or clear seasonal variations with the same sign. Overall, these seasonal fluctuations in the circulation-precipitation relationship can be attributed to factors influencing the types and mechanisms of precipitation, which can vary seasonally, such as atmospheric stability and thermodynamic feedback (Myoung and Nielsen-Gammon, 2010), or potential slight sampling bias in different seasons, as evidenced by the contrast between composite circulation maps for the same node in different seasons (Fig. S1).

**3.2 Moist circulation contributions to precipitation**

The above analyses show that strong seasonal variations could exist for either frequency or mean $P'$ in certain nodes due to different types of precipitation and factors other than large-scale circulation and moisture transport that contribute to precipitation. Therefore, assuming node-mean $P'$ as dynamic components calculated using the C2007 method may result in an incorrect representation of dynamic contribution and underestimation of its variability.

The second part of our analysis is to employ the SOMA method described in the method section to estimate the contribution of dynamics and thermodynamics to $P'$. We decompose the daily average $P'$ of certain regions (SGP, NGP, and SW) into the sum of a moist circulation-contributed component $P'_{SOMA}$ which is explained by the mid-tropospheric circulation and moisture transport, and a residual component $P'_{RES}$ which ideally should be independent

of circulation condition and related to thermodynamic processes such as land surface feedbacks. Further analysis shows that composite seasonal cycles of $P'_{SOMA}$ for different nodes are similar to that of $P'$ (Fig. 4) but with a narrower interquartile range (not shown). In contrast, $P'_{RES}$ does not show a clear season cycle for any node (not shown). This indicates the majority of precipitation variation at daily to synoptic scale can be mainly explained by circulation, and thermodynamic residual does not have a preference for certain nodes in any season. The $P'_{SOMA}$ calculated by our SOMA method overall is highly correlated to $P'$ in all seasons, indicating circulation and moisture transport conditions can explain a large portion of precipitation variability.

Fig. 5A shows the percentage of the variance of $P'$ explained by the $P'_{SOMA}$ ($R_{SOMA}^2$). NGP, which is located more to the north and more susceptible to frontal weather systems has the largest mean $R_{SOMA}^2$ of 60.5% for all year round, while SW and SGP have slightly smaller values of 56.3% and 53.6%, respectively. These three regions also show very clear seasonal variations of $R_{SOMA}^2$ as well: regions located more to the south like the SGP and SW, show a maximum of ~60% and ~70% in winter (December to January) when precipitation is more affected by frontal systems and a minimum of ~45% and ~40% in summer (July to August) when precipitation is mostly convectively driven and surface thermodynamic has more influences. On the other hand, the more northern region NGP shows continuous higher in the warm season (March to October); its seasonal variation of $R_{SOMA}^2$ shows two local minima, one in August (~55%) and the other in January (~50%), and two local maxima, one in April-May (~65%) and the other in October-November (~64%).

In addition to daily precipitation variability, we fit a simple linear regression model $P'_{SOMA} = \beta P'$ for each 91-day calendar window in each region to estimate the relative percentage dynamic contribution ($\beta$) to intensity of the precipitation anomaly. Fig. 5B shows the seasonal variation of $\beta$ for all samples at daily scale for each region. Overall, the percentage SOMA contributions in the SGP and SW show a similar seasonal cycle as the $R_{SOMA}^2$, with the highest $\beta$ (~69% and ~73%) in the cold season (December to January) and the lowest (~56% and ~50%) in the warm season (May to August), while the NGP has a mean $\beta$ of ~68% with a smaller range of seasonal variation (~65% to ~75%).

Fig. 6 further shows the historical evolution of 12-month moving average $P'$ and its moisture circulation contributed (SOMA) and residual (RES) components from 1950 to 2021 to demonstrate how they contribute to the longer-term variability of precipitation. Overall, $P'_{SOMA}$ matches better with $P'$ than $P'_{RES}$ does for all three regions, even for some $P'$ extremes during major drought and flooding events. Percentage variance explained by the SOMA component is 87.3%, 82.1%, and 82.8% for SGP, NGP, and SW, respectively, which are considerably higher than that explained by the residual (62.6%, 77.2%, and 65.5%). This suggests atmospheric circulation and moisture transport are the main controls for precipitation variations at interannual scale as well.

To further demonstrate what type of atmospheric circulation patterns are most related to regionally dry and wet anomalies, Fig. 7 ranks the nodes according to their mean seasonally cumulative $P'_{SOMA}$ and $P'_{RES}$ during four seasons, including January to March (JFM), April to June (AMJ), July to September (JAS), and October to December (OND). As these seasonal contributions are averaged over all years, many nodes show a clear tendency to be dynamically related to either dry or wet anomalies during a particular season in a region, but almost all of them have negligible residual components. Consistently through all three regions, the top four nodes that contribute most to dry anomalies in the cold season (JFM and OND) are A7, B7, C7, and D7, all located on the right side of the SOM map, featuring

either a west high-east low or high-dominant geopotential pattern over the CONUS with reduced IVT′ in both SGP and SW. On the other hand, for the wet anomalies, top contributing nodes are mostly located on the left or bottom side of the SOM map, featuring a west low-east high geopotential pattern favoring increased IVT′.

In the warm season (AMJ and JAS), top dry and wet nodes all change a bit for the three regions. For example, for the SGP, besides the high-dominant pattern D6 and D7, a few west low-east high patterns (C3, D2, D3) with increased IVT′ also contribute considerably to $P'_{SOMA}$ due to the increased lower atmospheric stability related (CIN′) related to them. For the SW, A1 becomes the top contributing node in its monsoon season due to the weak geopotential gradient there and the strong geopotential high on the west coast, all of which reduce the main moisture source, including that from the Gulf of Mexico and the Gulf of California.

### 3.3 Long-term changes in precipitation

The results from the previous section suggest that dynamic factors, specifically circulation and moisture transport, account for the majority of variance in precipitation anomalies from daily to interannual variability. Our next objective is to investigate whether there are long-term changes in precipitation and, if so, how these changes can be explained by circulation changes, which are mainly related to internal variability of the atmosphere and oceanic forcings. The PDO has been demonstrated as a main oceanic forcing that controls the multi-decadal variability of precipitation over North America, as PDO can generate atmospheric circulation patterns that favor dry or wet conditions depending on its phases (Lehner et al., 2018). Our period of analysis, spanning from 1950 to 2021, encompasses three periods with shifting PDO phases, including P1: 1950-1976 (mostly negative), P2: 1977-1998 (mostly positive), and P3: 1999-2021 (mostly negative).

Fig. 8 (1st column in each panel) shows the period mean $P'$ for different regions and contrasts them in three periods with different PDO phases. We can see that, overall, all three regions show a significant increase in $P'$ from P1 to P2, followed by a decrease from P2 to P3. The SOMA component $P'_{SOMA}$ (Fig. 8, 2nd column in each panel), though generally exhibits smaller amplitude anomalies, captures the multi-decadal changes of the $P'$ in the three periods (from 14% to 82% for all seasons, Table 2), especially for the SW (75% and 82% of the changes in P1 − P2 and P2 − P3).

To further determine how much of the dynamic contribution to the multi-decadal changes of $P'$ can be linked to PDO changes, we first investigate whether the moist circulation and precipitation anomalies associated with each node may be modulated by PDO phases. We assess whether the difference in frequency-weighted $P'_{SOMA}$ between positive and negative PDO phases is statistically significant through a Monte-Carlo test. Detailed steps are as following: a) for node $k$ in month $m$, calculate node frequency for positive PDO phase (monthly PDO index > 0.5) and negative phase (monthly PDO index < -0.5) for all years, i.e., $f_{PDO+}(k,m)$ and $f_{PDO-}(k,m)$, as well as average SOMA contributed $P'$, i.e., $P'_{SOMA,PDO+}(k,m)$ and $P'_{SOMA,PDO-}(k,m)$. b) the frequency weighted $P'_{SOMA}$ difference between positive and negative PDO phases for node $k$ in month $m$ can be calculated as $\Delta_{PDO}P'_{SOMA}(k,m) = f_{PDO+}(k,m) \times P'_{SOMA,PDO+}(k,m) - f_{PDO-}(k,m) \times P'_{SOMA,PDO-}(k,m)$, and overall PDO impact for node k can be represented by summing this difference across all months, i.e., $\Delta_{PDO}P'_{SOMA}(k) = \sum_{m=1}^{12}[\Delta_{PDO}P'_{SOMA}(k,m)]$. c) randomly shuffle the sequence of years for the PDO index data and recalculate the abovementioned statistics. d) repeat step c) many times (10000 here), if $\Delta_{PDO}P'_{SOMA}(k,m)$ is larger than 90% of all simulated values, then for node $k$ in month $m$, $P'_{SOMA}$ is significantly larger during positive

PDO phase than negative phase; in contrast, if $\Delta_{\text{PDO}}P'_{\text{SOMA}}(k,m)$ is smaller than 90% of all simulated values, then $P'_{\text{SOMA}}$ is considered significantly smaller during positive PDO phase. e) similar conclusions can be drawn for node k overall by assessing $\Delta_{\text{PDO}}P'_{\text{SOMA}}(k)$ instead of $\Delta_{\text{PDO}}P'_{\text{SOMA}}(k,m)$. The resulting nodes statistically related to PDO (for

each month and overall) are shown in Fig. 9. Additionally, the selection of a 0.5 threshold for PDO phases is mainly based on two considerations: 1) it has been commonly used in some prior studies (e.g., Hu and Guan 2018; Kiem 2003); 2) days or months categorized as positive or negative PDO phases using the 0.5 threshold constitute about 2/3 of the total samples (66.2%), striking a balance between inclusivity and specificity. Using a smaller threshold, such as 0.25, would result in a much higher percentage (81.8%) of samples categorized as positive or negative PDO-related,

leading to results that are less representative of the true PDO impact due to overinclusiveness. Conversely, a larger threshold, like 1.0, would yield fewer (40.6%) samples; although it is still feasible, which could introduce greater sampling uncertainty due to limited data availability. Results obtained using the 1.0 threshold can be found in the supplementary materials (Figs. S2-S4) for reference; overall, the 0.5 and 1.0 thresholds produce slightly different results, but these differences do not impact our subsequent discussion.

A significant $\Delta_{\text{PDO}}P'_{\text{SOMA}}$ value could be due to differences in node frequency, node mean $P'_{\text{SOMA}}$, or both, between the two PDO phases. Our result in Fig. 9 indicates that, for most nodes, PDO phases can have very different influences on precipitation through moist circulation conditions identified in this study, for different seasons and regions. For example, node A1, characterized by geopotential high anomalies along the west and east coasts and low anomalies over the central US, mirrors a Pacific North America (PNA)-like pattern, previously linked to Pacific variability and

its impact on precipitation in the western US and GP (e.g., Ciancarelli et al. 2014; Zhuang et al. 2021a). Interestingly, node A1 does not display a significantly higher frequency during the positive PDO phase compared to the negative phase (even lower in January to April; Fig. 10). However, it stands out as one of the nodes exhibiting the most substantial contrast in $P'_{\text{SOMA}}$ between the two PDO phases over the SGP. This contrast arises because the composite circulation pattern for node A1 during the positive PDO phase exhibits stronger negative $Z500'$ over the central US,

which can promote upward motion and is generally linked to reduced convection inhibition (Fig. S5), thereby leading to larger $P'_{\text{SOMA}}$.

Overall, when analyzing node frequency alone (Fig. 10), there are considerably more PDO-related nodes in the cold season from October to March (~13.5 nodes/month) than in the warm season from April to September (~6.8 nodes/month). This aligns with previous studies indicating that oceanic forcing has more influence on North American

precipitation variability in the cold season than in the warm season, largely due to increased atmospheric wave train activities excited by Pacific SST anomalies (Ciancarelli et al., 2014; Ropelewski and Halpert, 1986). However, when we account for the potential PDO impact on specific circulation patterns and corresponding within each node, the seasonal contrast in $\Delta_{\text{PDO}}P'_{\text{SOMA}}$ (Fig. 9) is reduced, with 6.5, 9.3, and 10.3 nodes/month in the cold season, and 5.7, 6.5, and 9.0 nodes/month in the warm season, are considered PDO related for the three regions, respectively.

Once the PDO-related nodes are identified, we accumulate $P'_{\text{SOMA}}$ only for days associated with significant $\Delta_{\text{PDO}}P'_{\text{SOMA}}$ as PDO-related circulation contributed $P'$ ($P'_{\text{SOMA-PDO}}$), and the result is shown in Fig. 8 (3rd column in each panel). We can see that although the PDO-related nodes we identified only constitute a small fraction of all nodes (about 22%, 28%, and 35% for SGP, NGP, and SW, respectively), $P'_{\text{SOMA-PDO}}$ shows very similar multi-decadal changes as the

total $P'_{SOMA}$. Table 1 summarizes the total and PDO-related dynamic contribution to the multi-decadal changes of $P'$. The dynamic contributions are highest for the SW (75% to 82%), with about 1/3 being PDO-related (32% to 39%). For the other two regions (SGP and NGP) with smaller dynamic contributions, $P'_{SOMA-PDO}$ also explains a relatively smaller portion of the total $P'$ change (mostly below 30%).

In addition to the change of mean $P'$, the change of $P'$ extremes is also of interest because these extremes are more impactful. Fig. 11 shows the probability density plot of percentile precipitation anomalies for $P'$ and $P'_{SOMA}$ in the three periods. During P1→P2, all three regions show a reduced probability of dry extremes and an increased probability of wet extremes. For the SGP and NGP, this decrease in dry extremes was more prominent in the warm season than in the cold season; the increase of wet extremes, though also varies seasonally, shows a larger change than dry extremes and can be found in all seasons. In the second period change P2→P3, as the PDO shifted back to the negative phase, all three regions showed an increase of dry extremes and a decrease of wet extremes, but it did not recover to the level as in P1, i.e., less dry extremes and more wet extremes than in P1. This is potentially because anthropogenic warming has increased the moisture in the atmosphere thus the recent P3 period favored wetter conditions and wet extremes more than the P1 period with the same negative PDO phase.

The moist circulation contribution $P'_{SOMA}$ has relatively flat curves at the drier side of the distribution (<10th percentile; defined as dry extremes), inconsistent with the $P'$ distribution, as dry extremes tend to be accompanied by stronger thermodynamic feedback; on the wetter side of the distribution (>90th percentile; defined as wet extremes), higher consistency between $P'$ and $P'_{SOMA}$ can be observed. Table 2 summarizes these observed changes in dry and wet extreme days and the corresponding total and PDO-related dynamic contribution. For the SGP and NGP, there is a ~5 d/yr decrease in dry extremes and ~10 d/yr increase in wet extremes during the earlier PDO shift (P1→P2, negative to positive); both total and PDO-related circulation contribution accounts for over 90% to the change in the SGP, while the percentages are much smaller for the NGP. However, during the recent PDO shift (P2→P3, positive to negative), the change of both dry and wet extremes in the SGP and NGP have considerably smaller amplitude compared to that during P1→P2, which could be related to the anthropogenic warming-induced increase in atmospheric moisture. SW has a greater change in extremes (-24 d/yr and +17 d/yr for dry and wet in P1→P2, +5 d/yr and -14 d/yr in P2→P3) compared to the other two regions. Moist circulation overall explains a significant portion of the changes, and most of the circulation contribution can be potentially related to PDO phase shift, especially for wet extremes; a lower circulation contribution for dry extremes (both overall and PDO-related circulation) may be due to the larger influence of thermodynamic feedback in the development of a dry event.

**4 Summary and Discussion**

SOM has been demonstrated as an effective way of identifying and visualizing large-scale circulation patterns related to various surface anomalies, such as precipitation and temperature (Liu and Weisberg, 2011). However, the SOM-based approach to quantify the contribution of anomalous circulation patterns on surface anomalies is not as reliable. On the other hand, flow analogue (including many of its variants) or the so-called dynamical adjustment technique has long been used as an important empirical approach to acquire or remove circulation anomaly variability, though it is not an effective way in terms of demonstrating what types of circulation is responsible for specific surface

anomalies or quantifying the contribution from a specific group of circulation. Building upon the previous studies related to these two methods, we develop a circulation contribution quantification approach based on both the two above-mentioned methods and use it to investigate the large-scale atmospheric circulation and moisture transport pattern related to the dry and wet precipitation anomalies over the U.S. Great Plains (SGP and NGP) and the Southwest (SW), as well as the moist circulation contribution to these anomalies, in terms of their daily to interannual variability

and long-term multi-decadal changes.

The first part of our results is a traditional SOM clustering analysis based on the large-scale atmospheric moist circulation anomaly pattern, including mid-tropospheric geopotential height ($Z500'$) and moisture transport ($IVT'$). We showed that many of these SOM nodes, i.e., types of circulation, have distinctive seasonal variations in terms of their frequencies of occurrence and regional precipitation anomalies related to them. This change of the circulation-

precipitation relationship is in part due to the seasonal difference in the main mechanisms producing the precipitation. For example, warm-season precipitation is mainly convectively driven so it is more related to convective energy and atmospheric stability, therefore nodes favoring wet conditions in the cold season which feature intense moisture transport (e.g., node D1, Fig. 4) could be linked to dry or neutral precipitation anomalies due to the less desirable convective statistics related to the same circulation condition.

We then went on to develop a SOM-Analogue approach and use the trained SOM to estimate the moist circulation contribution to precipitation anomalies over the SGP, NGP, and SW regions. Overall, moist circulation condition represented by $Z500$ and $IVT$ explains the majority of the variability and amplitude of $P'$, with a higher explained ratio in the cold season than in the warm season, and higher over the northern region (NGP) than the southern regions (SGP and SW). We also showed that the main nodes responsible for dry or wet anomalies could differ largely

depending on seasons and regions of interest.

The last part of the study further uses the partition results to estimate how much circulation can explain the multi-decadal shift of dry and wet mean $P'$ and extremes between 1950-1976, 1977-1998, and 1999-2021, and how much the PDO-related circulation can explain them given the overall distinctive PDO phase in each of the three periods. Our result suggests circulation explains the shift of mean $P'$ best in the SW (75-82%), with the PDO-related circulation

occupying a large portion (32-39%); while circulation may account for a greater proportion of variability and amplitude in daily $P'$ over the NGP compared to the other two more southern regions, it contributes less to the multi-decadal change in mean $P'$. The result is somewhat similar for the change of wet extreme days between the periods, with circulation and PDO-related circulation explaining a higher portion over the SW than over the SGP and NGP; circulation contribution drops significantly for change of dry extreme days than that of wet extreme days, indicating

thermodynamic factors such as land surface feedbacks and anthropogenic warming may play a more important role in the increase of dry extremes during the past decades.

This work demonstrates the effectiveness of the proposed SOM-Analogue in estimating circulation contribution to precipitation anomalies. This approach has the advantages of the C2007 SOM-based method, i.e., easy visualization of the circulation-precipitation anomaly relationship and quantification of contribution from different types of

circulation, yet is less sensitive to the choice of node numbers. It also has comparable performance as the constructed flow analogue method (Zhuang et al., 2021b) in terms of $R_{SOMA}^2$ (not shown), but $P'_{SOMA}$ calculated with SOM-

Analogue does not contain a thermodynamic residual as that in the analogue method. However, this is still a statistical method with certain assumptions, so limitations still exist. For example, it could become less reliable when fewer samples are presented; both analogue and SOM-Analogue approaches all assume a constant circulation-precipitation anomaly relationship which might not hold true in a changing climate. Furthermore, the identified PDO-related SOM nodes and $P'$ are likely also modulated by tropical SST variability. This is due to the well-established understanding that ENSO and PDO can generate similar atmospheric and oceanic anomaly patterns (e.g., Hu and Huang, 2009). Additionally, other internal climate variability modes, such as the Atlantic Multidecadal Oscillation (AMO; e.g., Hu et al., 2011), North Atlantic Oscillation (NAO; e.g., Whan and Zwiers, 2017), and Interdecadal Pacific Oscillation (IPO; e.g., Dai, 2013), can also influence these patterns. The SOM circulation patterns defined by Z500′ and IVT′ simply represent regional-scale manifestation of larger-scale variability simultaneously influenced by multiple internal climate modes. Attempting to isolate the individual contribution of these modes using statistical methods with limited data, such as SOM or SOMA, can be a complex challenge. Therefore, our results related to PDO contribution serve as a preliminary starting point which demonstrates the combined contribution of the SOM node patterns statistically linked to the PDO. To gain a more detailed quantification of the PDO's influence excluding the effects of other climate variability modes, further research integrating both observational data and climate model output is needed.

**Code Availability**

The Matlab SOM Toolbox, used for training the SOM in this study, is available at http://www.cis.hut.fi/somtoolbox/. Additional code developed for generating the results of this study can be provided by the corresponding author upon reasonable request.

**Data Availability**

All data used in this study are publicly accessible. ERA5 data can be obtained via the Climate Data Store (https://cds.climate.copernicus.eu/). CPC Precipitation data can be accessed through the NOAA Physical Sciences Laboratory (PSL)'s FTP server at ftp.cdc.noaa.gov. The NCEI PDO index is available at https://www.ncei.noaa.gov/access/monitoring/pdo.

**Author Contribution**

YZ and RF designed the research. YZ developed the methodology, performed formal analysis, and wrote the initial draft. RF acquired funding support and managed the project. YZ and RF reviewed and edited the draft.

**Competing Interests**

The contact author has declared that none of the authors has any competing interests.

**Acknowledgements**

This study was supported by funding from the National Oceanic and Atmospheric Administration-Climate Program Office (NOAA-CPO), Modeling, Analysis, Predictions, and Projections (MAPP) Program (NA20OAR4310426), and the National Science Foundation (NSF) Physical and Dynamic Meteorology (PDM) Program (AGS-2214697).

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

**Table 1: Long-term mean precipitation anomaly changes across three periods with shifting PDO phases (P1: 1950-1976,**

**PDO-; P2: 1977-1998, PDO+; P3: 1999-2021, PDO-) and the total and PDO-related moist circulation (SOMA) contribution.**

| Region | Observed change (mm/d) | | SOMA contribution (%) | | | |
|---|---|---|---|---|---|---|
| | | | Total | | PDO-related | |
| | P1 → P2 | P2 → P3 | P1 → P2 | P2 → P3 | P1 → P2 | P2 → P3 |
| SGP | +0.260 | -0.087 | 62.4 | 28.4 | 23.3 | 24.9 |
| NGP | +0.195 | -0.043 | 36.4 | 14.4 | 6.9 | 28.0 |
| SW | +0.211 | -0.133 | 74.9 | 81.9 | 32.1 | 39.0 |

**Table 2: Same as Table 1, but for change of extreme dry days ($P'$ below 10th percentile) and wet days ($P'$ above 90th percentile).**

| Extreme type | Region | Observed change (d/yr) | | Moist circulation contribution (%) | | | |
|---|---|---|---|---|---|---|---|
| | | | | Total | | PDO-related | |
| | | P1 → P2 | P2 → P3 | P1 → P2 | P2 → P3 | P1 → P2 | P2 → P3 |
| Dry | SGP | -4.8 | +0.1 | 93.1 | 940.6 | 117.3 | 994.7 |
| | NGP | -6.1 | -3.3 | 10.3 | 18.2 | -19.3 | 70.5 |
| | SW | -24.2 | +5.2 | 23.8 | 50.8 | 11.6 | 76.7 |
| Wet | SGP | +9.7 | +1.2 | 93.8 | -59.3 | 130.7 | -196.3 |
| | NGP | +11.0 | -4.7 | 59.9 | 54.0 | 29.2 | 121.3 |
| | SW | +16.8 | -14.2 | 115.27 | 94.7 | 150.7 | 130.0 |

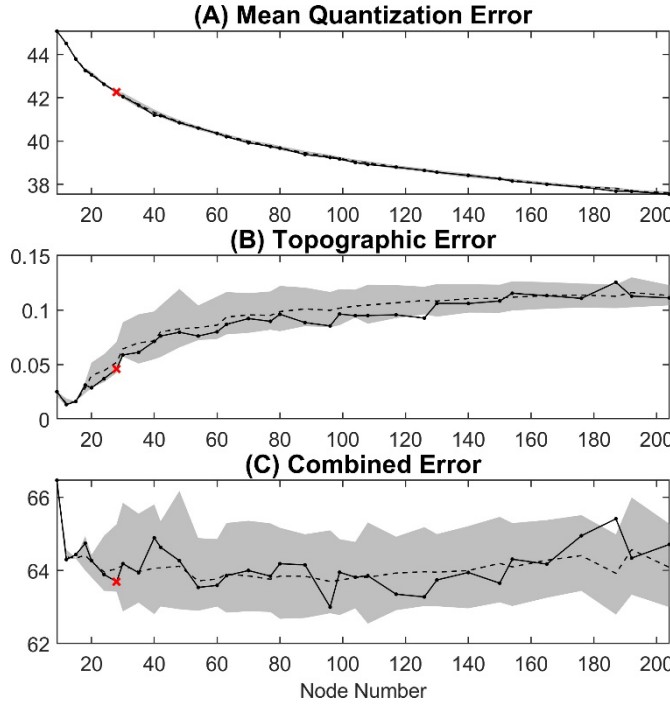

**Figure 1: (A) Quantization error (QE), (B) topographic error (TE), and (C) combined error (CE) of SOM schemes with different node numbers ($N$). The red cross denotes the one ($N = 28$) we selected in this study. Solid lines represent the errors**

**of the SOMs trained with linear initialization, while the shaded areas represent the range of errors within the middle 95%**

**of the distribution from the 100 SOMs trained with random initialization for each node number setting, and dashed lines are their averages.**

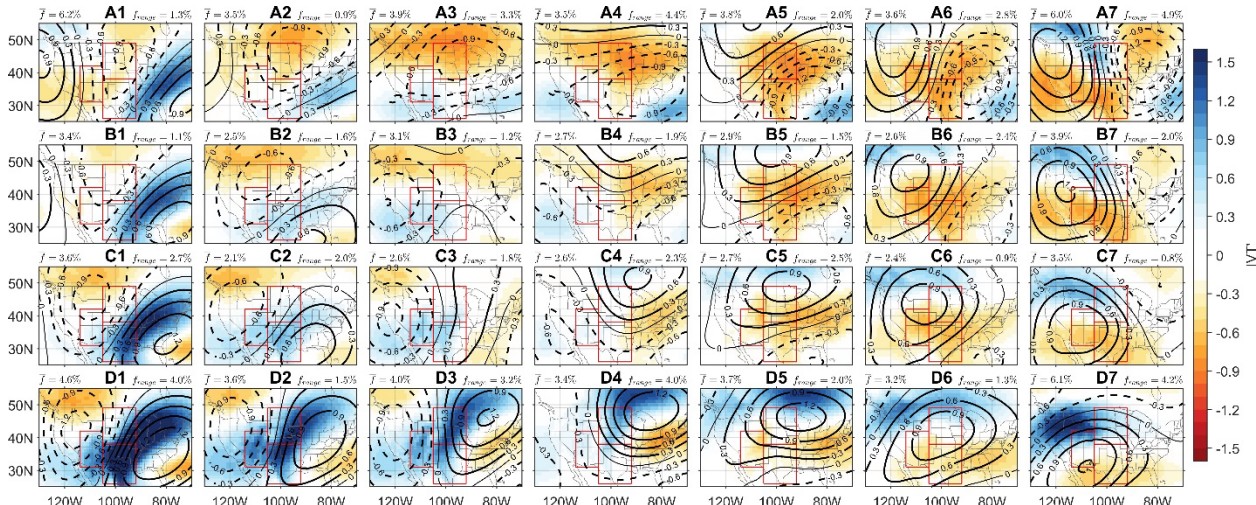

**Figure 2: Composite standardized Z500′ (contours; solid and dashed lines are for positive and negative values, respectively) and standardized IVT′ (shades) for each node in the 7×4 SOM. Mean frequency ($\bar{f}$) and seasonality ($f_{range}$, i.e., the difference between maximum and minimum seasonal frequencies as shown in Fig. 3, a larger value indicates stronger seasonal variation) of each node is labeled in each panel.**

**Figure 3: Seasonal variation of frequency for SOM nodes. For each node in each panel, the frequency at a certain calendar day represents the node frequency during a 91-day calendar window centered on that calendar day for all years (1950-2021).**

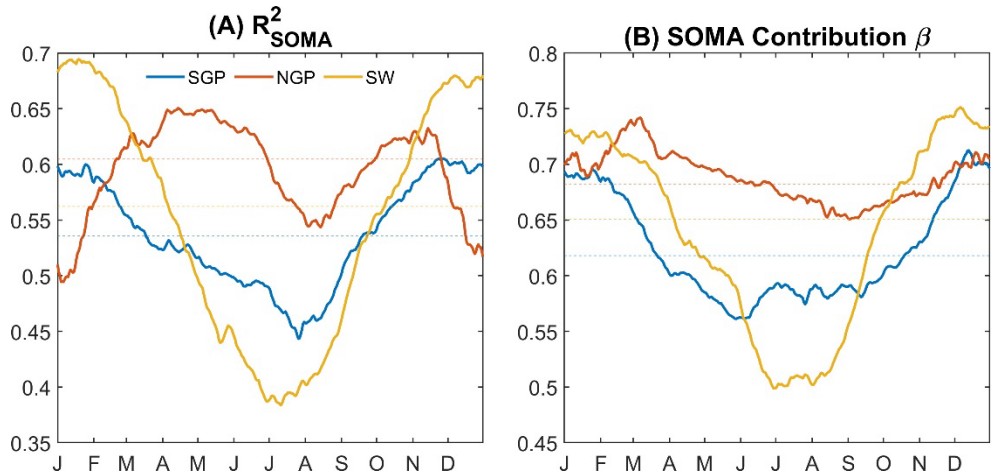

**Figure 4: Seasonal variation of mean *P′* averaged over each region (blue for SGP, red for NGP, and yellow for SW) for each SOM node. For each node in each panel, the mean *P′* at a certain calendar day represents the *P′* of that node during a 91-day calendar window centered on that calendar day for all years (1950-2021). The shade represents the interquartile range of all daily *P′* used to calculate the mean.**

**Figure 5: (A) Seasonal variation of percentage *P′* variance explained by *P′*$_{SOMA}$ ($R_{SOMA}^2$) for the three regions; each data point is calculated within a 91-day calendar window. Thin dashed lines represent the value for all year round. (B) Same as (A), but for seasonal variation of percentage moist circulation (SOMA) contribution determined by fitting a simple linear regression model between *P′*$_{SOMA}$ and *P′* during a 91-day calendar window (*P′*$_{SOMA}$ = $\beta$*P′*).**

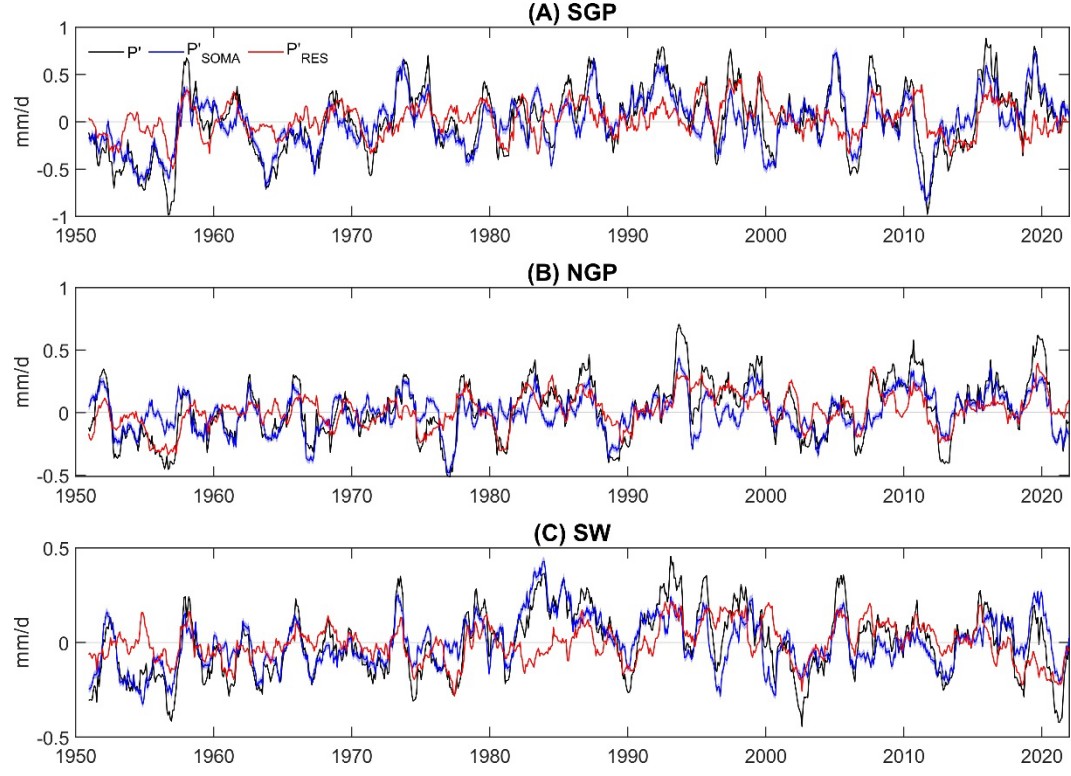

**Figure 6: 12-month moving average *P′* (assigned to the last month of a 12-month period) for the three regions. Black, blue, and red lines represent the total anomaly *P′*, the moist circulation contributed component *P′*SOMA, and the residual component *P′*RES, i.e., *P′* − *P′*SOMA.**

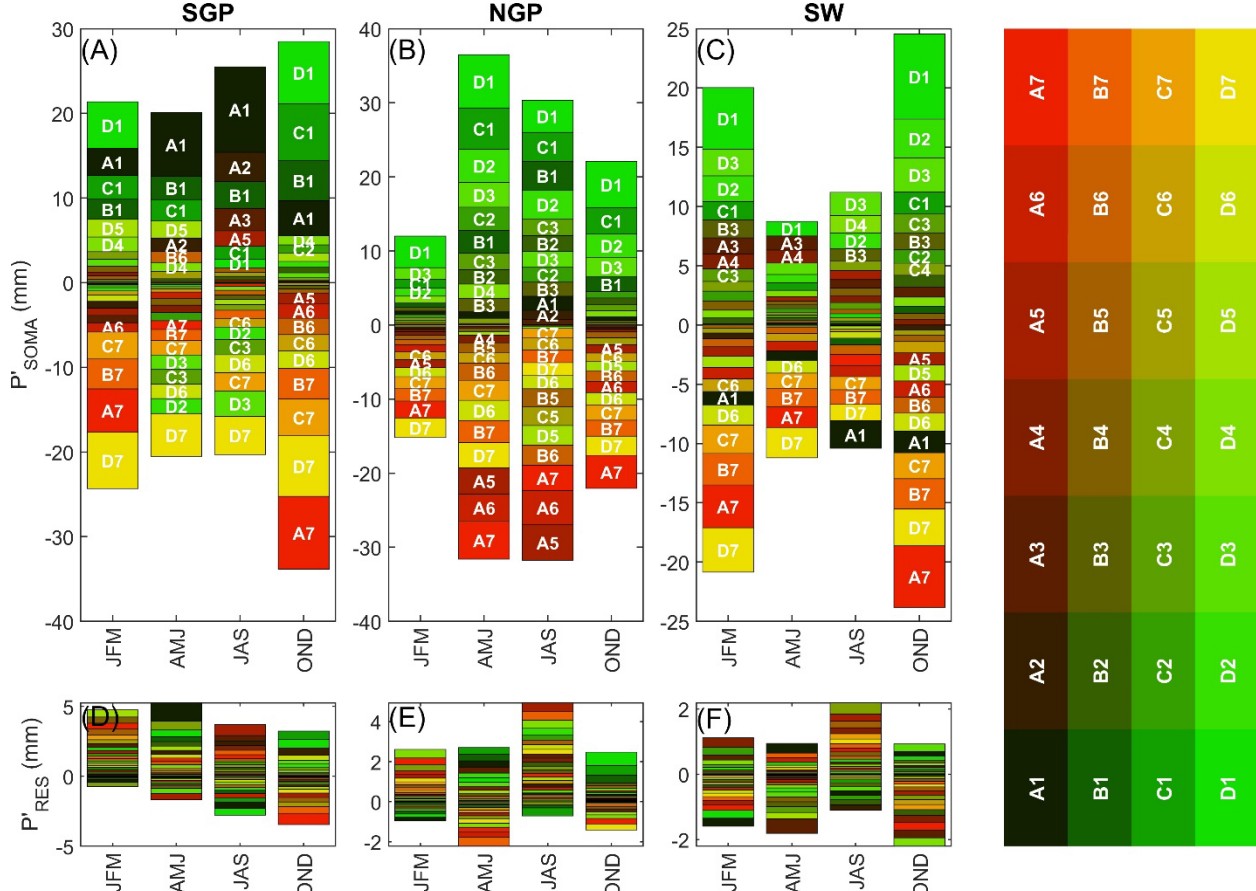

**Figure 7: Average moist circulation contribution (SOMA; A-C) and residual (RES; D-F) to $P'$ from each node in each three-month season (Jan-Mar, Apr-Jun, Jul-Sep, and Oct-Dec) over each region (SGP, NGP, SW). The colors of nodes are assigned in such a manner that adjacent nodes exhibit greater similarity in color compared to nodes that are farther apart.**

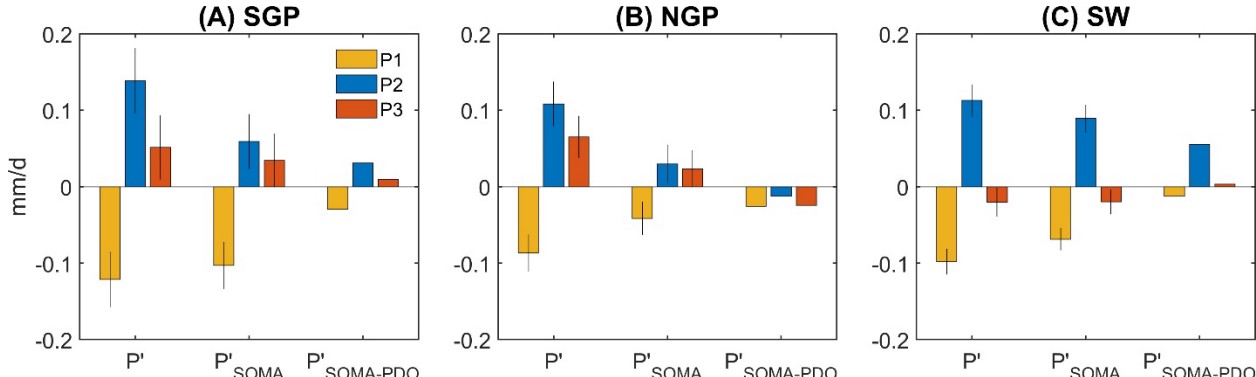

**Figure 8: Mean $P'$, $P'_{SOMA}$, and $P'_{SOMA-PDO}$ during three periods with shifting PDO phases (P1: 1950-1976; P2: 1977-1998; P3: 1999-2021) in the three regions: (A) SGP, (B) NGP, and (C) SW. Error bars in (A) and (B) represent the 95% confidence intervals of the mean $P'$ and $P'_{SOMA}$.**

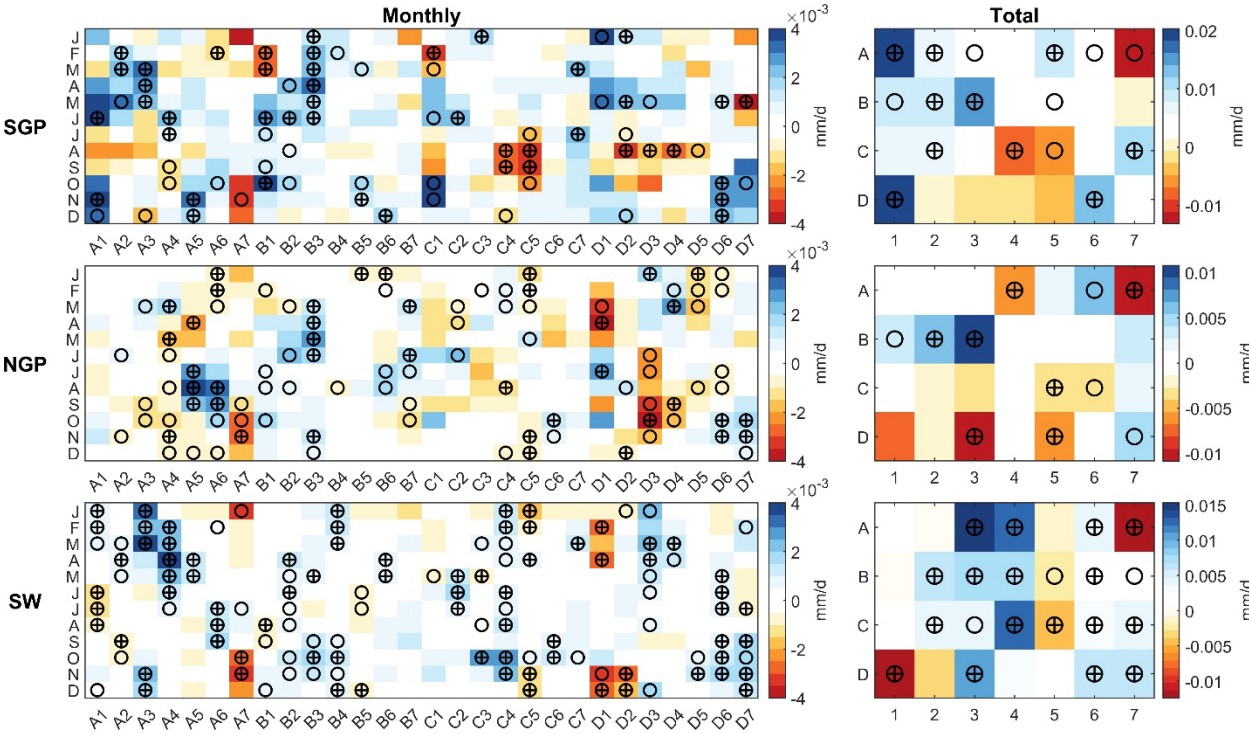

**Figure 9: Frequency weighted $P'_{SOMA}$ difference between positive and negative PDO phases (monthly PDO index >0.5 and <-0.5) for different nodes in different months. For each row or month, nodes marked with a circle (with a plus sign) indicate the frequency differences are significant positive or negative at 0.10 (0.05) level, and they can be considered PDO-related nodes for that month.**

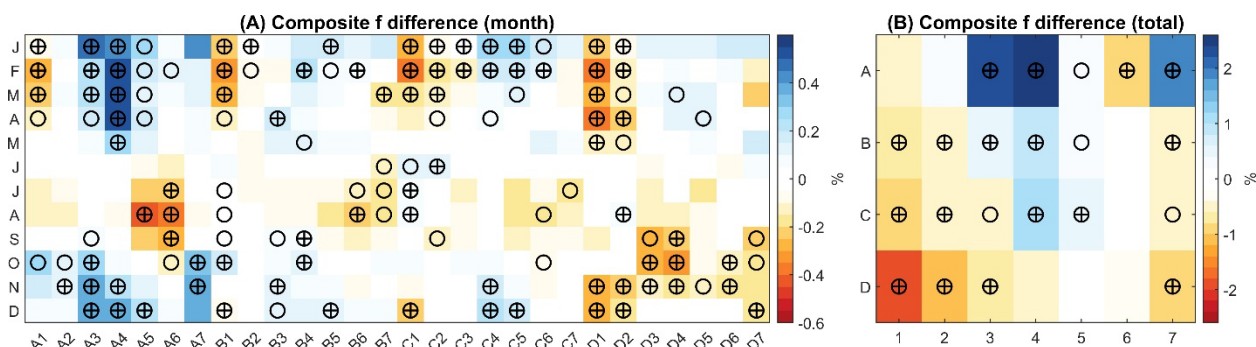

**Figure 10: Same as Fig. 9, but for frequency difference.**

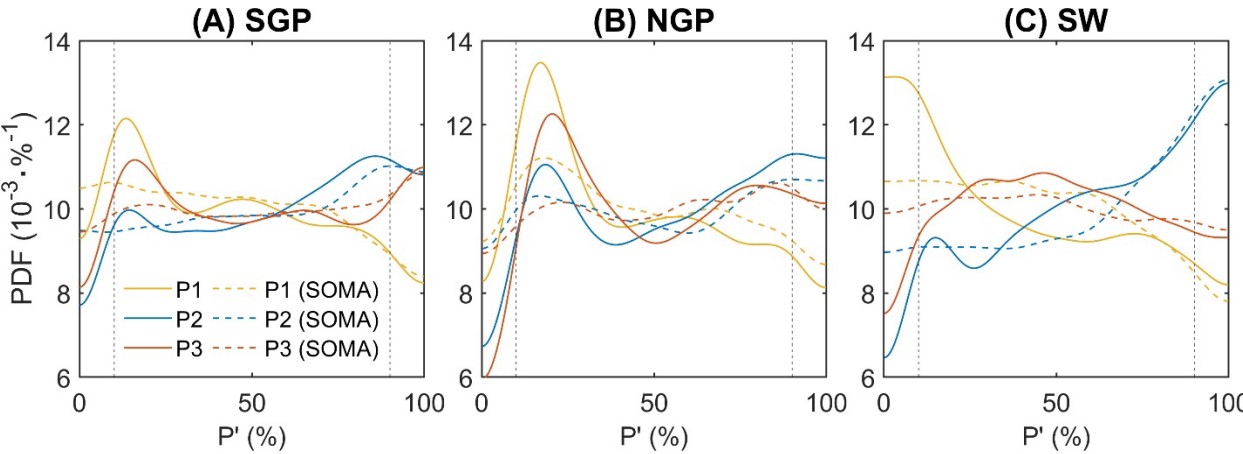

**Figure 11: Kernel density plot of percentile *P′* and *P′*SOMA during the three periods with shifting PDO phases (P1: 1950-1976, P2: 1977-1998, P3: 1999-2021) in three regions: (A) SGP, (B) NGP, and (C) SW. Solid and dashed lines represent percentile *P′* and *P′*SOMA, respectively.**