# Peer review of "Quantifying Contribution of Atmospheric Circulation to Precipitation Variability and Changes in the U.S. Great Plains and Southwest Using Self Organizing Map – Analogue"

_EGUsphere, 2023_

## Author Comment (AC1)

**Reply to Reviewer's Comments**

**Reviewer 1:**

Review of: Quantifying Contribution of Atmospheric Circulation to Precipitation Variability and Changes in the U.S. Great Plains and Southwest Using Self Organizing Map – Analogue

OVERALL COMMENTS:

This is a really interesting paper. I think this is a valuable contribution. I am a big proponent of the use of classification techniques (and SOMs) in helping to analyze and visualize the complex relationships between atmosphere and surface impacts – and this is an excellent example. However, I have some semi-major and minor comments that I detail below, that I would like the authors to address and consider, prior to publication. Also, this manuscript might need some slight language editing.

MAJOR COMMENTS:

I think the dynamic vs. thermodynamic dichotomy is a bit misleading. This is because z500 and IVT are going to have **thermally-related variability inherent to them**, right? While you only explicitly include z500 heights and IVT on each day into the classification, that does not necessarily mean that many/all other environmental variables (e.g. 2m temperature, 850t, 2m dew points, 925winds, SLP, and everything else) are not also 'indirectly' playing a role in classifying a day's weather. That is to say, classification is implicitly wholistic – it categorizes the *wholistic/synergistic* environment over a particular time period (herein, a day). While this is not an issue in most research using classification (and is arguably a benefit of using classification in applied research), herein, when you are trying to use classification to de-couple the dynamics portion from thermodynamics, I think it is problematic. I just cannot de-couple this in this manner… To me, what you are actually calculating here is not dynamic vs. thermodynamic contributions, but rather the variability that can be accounted for using this categorization (i.e. this SOMA-based model) and the residual variability that cannot – and both the SOM and the residual contain both dynamic and thermodynamic contributions within them. That is, you do not have P'dyn vs. P'the, but rather more like **P'SOMA vs. P'residual**. In this sense, there is nothing inherently wrong with what you did, just the way you interpreted it, and the dynamic vs. thermodynamic is a misnomer. But, perhaps I can be convinced otherwise.

Thank you very much for this comment/suggestion! We agree that dynamics and thermodynamics are not two independent processes, and precipitation corresponding to certain Z500/IVT pattern types have thermally related components. We have taken your suggestions and revised the decomposed terms from $P'_{dyn}$ / $P'_{the}$ to $P'_{SOMA}$ / $P'_{RES}$. In addition, we added some texts in the method section to reflect this consideration.

Line 193-199: "It's worth noting that attempting to separate the actual dynamic and thermodynamic components solely through a circulation clustering approach like SOM can be

challenging. Each type of circulation, as represented by SOM nodes, inherently encompasses thermodynamic responses. Therefore, the distinction between "dynamic" and "thermodynamic" components can be ambiguous when using these terms. Consequently, we prefer to refer to the precipitation influenced by moist circulation patterns involving Z500 and IVT as $P'_{SOMA}$ instead of $P'_{dyn}$. This emphasizes that our results regarding circulation contributions are contingent on our chosen set of circulation variables."

INDIVIDUAL COMMENTS:

Line 100-102: Need a bit of clarity here… the daily standardized anomaly is applied to the pentad moving average filter? Or the opposite? Or are these two things done separately?

Yes, we calculated the standardized anomaly after applying the 5-day moving average filter. We clarify this by revising Line 101-103: "To mitigate high-frequency synoptic noise, we employ a simple 5-day moving average filter to both Z500 and IVT. Subsequently, we calculate the daily standardized anomaly (Z500' and IVT') using the 5-day moving average-filtered data relative to the 1950-1999 climatology.".

Line 102: Why 1950 to 1999 for the climo? Why not the entire period, or the most recent 30-year climate normal period? Or a period that ends in the present (2021)?

We added a sentence at Line 103-105 to explain the choice of climatology:
"We select the 1950-1999 period as the climatology reference to maximize the utilization of available data for this study; furthermore, this period, which predates the significant warming trend, typically serves as a robust baseline with less climate change impact."

Line 131: The way this is written is still slightly confusing in terms of how many dimensions you actually have here. It is a 2-dimensional data matrix with a size of 3782-by-26,280 (with perhaps a few leap-days in there), correct?

Thanks for pointing this out. We revise Line 132-133:

"Specifically for this study, $N_{lon}$ = 61, $N_{lat}$ =31, and $N_t$ = 365 days/year × 72 years (leap days removed for simplicity), so the input size is (3782, 26280)."

Line 160: While I am fine with the way you did this, just as a note, if trained using a batch process, then slightly different input vectors might be used and thus, result in different final clustering solutions, even with all the other 'settings' the same. So, you might want to run each node number multiple (10-20) times, and then average their QE, TE and CE.

Thank you very much for the comments. The SOM training was done with linear initialization so input vectors are fixed. In order to quantify the influence of initialization on the outcome, we adopt your suggestion and run each node number setting for an additional 100 times with random initialization setting, and plot their range in Fig. 1 (new Fig. 1 attached here). We still present the result from our original N=28 setting with linear initialization to facilitate replicability of the result.

Relevant texts are added and attached here as well:

Line 159-165: "During the initialization before the training, we employ the default setting, i.e., linear initialization, where weight vectors are initialized in a linear manner along the subspace defined by the two principal eigenvectors of the input dataset. This choice is made to facilitate the reproducibility of our results. While SOM is generally robust to initialization, slight variations in outcomes may occur when using random initialization, where the weight vectors are initialized with random small values. To assess the impact of initialization on SOM error metrics, we also conduct 100 additional SOM training with random initialization for each node number setting and analyze the range of their error metrics."

[Figure]

Figure 1: (A) Quantization error (QE), (B) topographic error (TE), and (C) combined error (CE) of SOM schemes with different node numbers ($N$). The red cross denotes the one ($N$ = 28) we select in this study. Solid lines represent error of the SOM trained with linear initialization, while the shaded areas represent the range of errors within the middle 95% of the distribution from the 100 SOMs trained with random initialization for each node number setting, and dashed lines are their averages.

Figure 2: It would be nice if you could incorporate the seasonality of frequency of each atmospheric pattern into this graphic instead of having a separate figure 3.

Thanks for the suggestion. However, we found it very difficult to combine Figs. 2 and 3 without making it look visually busy. Instead, we added the mean frequency and seasonal frequency range numbers in each subplot of Fig. 2 (attached below) to somehow incorporate the frequency seasonality of each pattern. We kept Fig. 3 as it is to provide more detailed frequency information in addition to Fig. 2.

[Figure]

**Figure 2: Composite standardized Z500' (contours; solid and dashed lines are for positive and negative values, respectively) and standardized IVT' (shades) for each node in the 7×4 SOM. Mean frequency ($\bar{f}$) and seasonality ($f_{range}$, i.e., difference between maximum and minimum seasonal frequencies as shown in Fig. 3, large value indicate stronger seasonal variation) of each node is labeled in each panel.**

Figure 3: I think the y-axis on these should be identical, so we can tell which ones are more/less frequent overall.

Thanks, same y-axis is used in the revised figure.

Figure 4: Again, I think the y-axis on these should all be identical.

Revised.

Line 226 and Figure 5: I rarely see CAPE and CIN in standardized values, but rather in their more-traditional units. I think you need to be careful here, as for example, a lower than average CAPE for a location that has pretty high average CAPE normally, might still mean that the atmosphere is pretty unstable. Also, are these values deseasonalized?

We agree that lower than average CAPE for a location/season could still mean a higher absolute CAPE value compared to other locations/seasons. We revise clarify this:

Line 264-270: "Additionally, the composite CAPE' and CINi' map for node D1 (Fig. S1) shows relatively smaller CAPE' and a more stable lower troposphere (negative CINi') compared to other nodes in the warm season (May to July), suppressing convective development thus limiting precipitation regardless of the strong moisture transport, whereas in the cold season (November to January), CAPE' and CINi' shows larger positive values associated with more precipitation compared to other nodes. However, it's important to note that the above comparison is within the same season, as node D1 in the warm season still corresponds to higher traditional CAPE and CINi values (not anomalies), as well as less total precipitation, compared to the cold season."

And yes, these values are deseasonalized (climatology removed). Relevant texts are at:

Line 121-123: "Both CAPE and CINi data are processed the same way as circulation data to acquire their standardized anomalies (CAPE' and CINi') for further analyses."

Line 230: D1 is perhaps suppressing convective development, but convective *initiation* (i.e. the 'triggering lifting mechanism') is as somewhat separate ingredient.

Thank you for pointing out this. We remove "initiation" and keep "development" in the revision (Line 266).

Line 235: the soil moisture thing comes out of no-where…. I don't disagree, but, I do suggest a citation for this.

Thank you for pointing out this. We revise this to use "thermodynamic feedback" instead of "soil moisture feedback" to avoid confusion. A references is also added.

Line 272-275: "Overall, these seasonal fluctuations in the circulation-precipitation relationship can be attributed to factors influencing the types and mechanisms of precipitation, which can vary seasonally, such as atmospheric stability and thermodynamic feedback (Myoung and Nielsen-Gammon, 2010), or potential slight sampling bias in different seasons, as evidenced by the contrast between composite circulation maps for the same node in different seasons (Fig. S1)."

Line 274-275: I think you need to be careful with how this sentence is worded. I agree that the dynamic factors are likely the major contributors, but that statement is predicated upon how you specifically defined "dynamics" herein (with z500 and IVT SOMs). If you had chosen different variables to represent "dynamics" (e.g. SLP) would this statement still hold? Would the statement be weaker? Would it be stronger?

To address your major comment which is related to this comment as well, we change the use of "dynamic contribution" to "moist circulation contribution" (as suggested by the other reviewer) or "SOMA contribution", in order to reflect that these statements are based on the circulation variables (Z500+IVT) we used.

To answer your other question about using other dynamic variables, we did find that using SLP instead of Z500 or Z500+IVT would yield a smaller percentage of explained variance. We briefly mentioned this in the first few sentences in the data section (attached below) but did not show a more detailed comparison for simplicity.

Line 90-93: "In previous studies related to SOM and analogue, large-scale circulation is generally represented by mean sea level pressure (SLP) or geopotential height at 500 hPa (Z500). Here, we choose Z500 over SLP as our experiments have demonstrated analogues derived from Z500 show greater similarity in synoptic variability with observed surface anomalies and yield smaller residuals compared to analogues derived from SLP (Zhuang et al., 2021b)."

Figure 8: Very interesting figure! I like it. However, why are the types color-coded the way they are? Is there a reason? It looks roughly like wet is green, and dry is red/orange/yellow, but is there a specific method for this?

Thank you. We added a sentence in the caption to explain this: "The colors of nodes are assigned in such a manner that adjacent nodes exhibit greater similarity in color compared to nodes that are farther apart.", as well as a new panel to show the gradual change in colors across nodes (new Fig. 7 attached below).

[Figure]

**Figure 7: Average moist circulation contribution (SOMA; A-C) and residual (RES; D-F) to *P'* from each node in each three-month season (Jan-Mar, Apr-Jun, Jul-Sep, and Oct-Dec) over each region (SGP, NGP, SW). The colors of nodes are assigned in such a manner that adjacent nodes exhibit greater similarity in color compared to nodes that are farther apart.**

Figure 9: Again, these subplots should have the same y-axis.

Same y-axis is applied to all subplots in the revision.

Line 314: "… larger or smaller than 90%…" - wouldn't that be ALL of them?

No, we mean when is larger than >90% of all (so smaller than <10% of them), or smaller than >90% of them (so larger than <10% of them).

We revise Line 356-359 to clarify this: "d) repeat step c) for many times (10000 here), if $\Delta_{PDO}P'_{SOMA}(k,m)$ is larger than 90% of all simulated values, then for node $k$ in month $m$, $P'_{SOMA}$ is significantly larger during positive PDO phase than negative phase; in contrast, if $\Delta_{PDO}P'_{SOMA}(k,m)$ is smaller than 90% of all simulated values, then $P'_{SOMA}$ is considered significantly smaller during positive PDO phase."

Line 316: yes, but this is partly because PDO is not as often >0.5 or <-0.5 during these months, and thus, less sample size, right? And why the +/- 0.5 thresholds... why not 1.0, or 0.25??

We added Line 361-369 to explain the choice of threshold.

"Additionally, the selection of a 0.5 threshold for PDO phases is mainly based on two considerations: 1) it has been commonly used in some prior studies (e.g., Hu and Guan 2018; Kiem 2003); 2) days or months categorized as positive or negative PDO phases using the 0.5 threshold constitute about 2/3 of the total samples (66.2%), striking a balance between inclusivity and specificity. Using a smaller threshold, such as 0.25, would result in a much higher percentage (81.8%) of samples categorized as positive or negative PDO-related, leading to results that are less representative of the true PDO impact due to overinclusiveness. Conversely, a larger threshold, like 1.0, would yield fewer (40.6%) samples; although it is still feasible, which could introduce greater sampling uncertainty due to limited data availability. Results obtained using the 1.0 threshold can be found in the supplementary materials (Figs. S2-S4) for reference; overall, the 0.5 and 1.0 thresholds produce slightly different results, but these differences do not impact our subsequent discussion."

Line 321: Perhaps I am missing something here, but, just because these nodes have been found to be related to PDO, doesn't necessarily mean that this represents the "PDO-related dynamic contribution" to P' ... moreover the way that you have constructed this, makes it impossible for PDO to contribute MORE than the SOM-based dynamic component. That is, it could very well be that PDO is a more-dominant factor (more dominant than the SOM patterns), but this methodology would not allow for that. I think there is a better way to tease out how PDO is contributing to P' in these areas. At best, perhaps you can say that this is showing the combined contribution of the SOM-patterns that have been found to be most influenced by the PDO.

Also, who is to say that this represents PDO-only related contributions to P'? What if you did this for AMO and AMO phases?? Or the IPO? In many ways, the SOM patterns you define are simply the regional-scale manifestation of hemispheric- to global-scale variability in the multiple internal climate oscillations occurring simultaneously (PDO/AMO/IPO/ENSO and that of all other teleconnections/oscillations at various periodicities).

Thank you for pointing out these limitations. And we agree that it would be impossible to disentangle the PDO contributions from various climate modes simply by using our current form of methodology. We added discussion about these limitations in the discussion section (attached below). Hope this can address your concerns.

Line 471-482: "Furthermore, the identified PDO-related SOM nodes and $P'$ is likely also modulated by tropical SST variability. This is due to the well-established understanding that ENSO and PDO can generate similar atmospheric and oceanic anomaly patterns (e.g., Hu and Huang, 2009). Additionally, other internal climate variability modes, such as the Atlantic Multidecadal Oscillation (AMO; e.g., Hu et al., 2011), North Atlantic Oscillation (NAO; e.g., Whan and Zwiers, 2017), and Interdecadal Pacific Oscillation (IPO; e.g., Dai, 2013), can also influence these patterns. The SOM circulation patterns defined by Z500' and IVT' simply represent regional-scale manifestation of larger-scale variability simultaneously influenced by multiple internal climate modes. Attempting to isolate the individual contribution of these modes using statistical methods with limited data, such as SOM or SOMA, can be a complex challenge. Therefore, our results related to PDO contribution serves as a preliminary starting point which demonstrate the combined contribution of the SOM node patterns statistically linked to the PDO. To gain a more detail quantification of the PDO's influence excluding the effects of other climate variability modes, further research integrating both observational data and climate model output is needed."

---

## Author Comment (AC2)

**Reply to Reviewer's Comments**

**Reviewer 2:**

GENERAL COMMENTS:

Overall, this paper can be great contribution to our science community. The authors come up with the novel approach for quantifying the contribution of atmospheric circulation to the recent precipitation variability in US. They combine two statistical techniques - Self Organizing Map (SOM) and circulation analogue - to effectively visualizing circulation pattern as in SOM and quantifying the dynamical contribution as exact as in circulation analogue. Using this new SOM-Analogue (SOMA) approach, they investigate the link between circulation patterns and daily precipitation anomalies in different seasons and PDO phases. The results provide the valuable insight how the specific circulation pattern leads to the regional precipitation anomalies in US. However, I think there are some major issues in the development of new SOMA approach and the analysis regarding PDO-related contributions. I would suggest the authors to address below major/minor comments before the publication.

MAJOR (SPECIFIC) COMMENTS:

Decomposition of dynamic and thermodynamic contribution

The variables Z500 and IVT are used for elucidating the dynamic contribution to the precipitation anomalies. However, IVT is multiplication of wind vector (dynamic variable) and moisture (thermodynamic variable), so the SOMA from IVT would also contain the thermodynamic contribution. To decompose the dynamic and thermodynamic contribution, I think dynamic variables (e.g., velocity potential) has to be used instead of IVT. Or, rather than focusing on the decomposition of dynamic/thermodynamic contribution, the authors can express the same results as the contribution from the moist circulation and the residual. In my opinion, the interesting science in this study is to characterize the circulation pattern responsible for the regional precipitation anomalies, and the exact decomposition of thermodynamic and dynamic contribution is kind of secondary interest compared to that.

If the authors want to decompose the thermodynamic/dynamic contribution, then I think new variable should be used for SOMA instead of IVT. Or, authors can express the same results as the contribution from the moist circulation and residual, with revising rest of the manuscript accordingly.

Thank you very much for this helpful comment! The other reviewer also suggested that a decomposition based on wholistic contribution from the SOMA circulation vs. residual is more appropriate. We took your advice and changed $P'_{dyn}$ and $P'_{the}$ to $P'_{SOMA}$ and $P'_{RES}$. In the texts, we also changed terms like "dynamic contribution" to "moist circulation contribution" or

"SOMA contribution". In addition, we added some texts in the method section to reflect this consideration.

Line 193-199: "It's worth noting that attempting to separate the actual dynamic and thermodynamic components solely through a circulation clustering approach like SOM can be challenging. Each type of circulation, as represented by SOM nodes, inherently encompasses thermodynamic responses. Therefore, the distinction between "dynamic" and "thermodynamic" components can be ambiguous when using these terms. Consequently, we prefer to refer to the precipitation influenced by moist circulation patterns involving Z500 and IVT as $P'_{SOMA}$ instead of $P'_{dyn}$. This emphasizes that our results regarding circulation contributions are contingent on our chosen set of circulation variables."

Issues regarding the data and SOMA approach

First, is the observational data used for SOMA **detrended or high-pass filtered**? In previous studies using circulation analogue (Deser et al. 2016; Lehner et al. 2018), the dynamic contribution is found after the observational data is detrended or high-pass filtered. This is for eliminating the forced thermodynamic contribution in the timeseries and to focus on the dynamic contribution. In this manuscript, it seems there are no mentioning about such data processing. I think the detrend is needed if it is not done, and it should be mentioned if it is done. Or, you may show that such data processing doesn't matter to your results.

Yes, all data including circulation and precipitation are high-pass filtered with a simple 5-day moving average. For Z500, global mean Z500' are subtracted from original data prior to calculating anomalies to account for thermal expansion of the atmosphere due to warming following prior studies, so there is no apparent trend in Z500 data. We did not use the traditional detrending approach (remove linear trend) because there could be multidecadal variability (both circulation and precipitation) linked to internal variability which could sometimes falsely appear as a weak linear trend. These were mentioned in the data section (2.1) and we revise those sentences (attached below) to make them clearer to the authors the data is high-pass filtered and detrended.

Line 98-102: " To account for the thermal expansion of the warming atmosphere, we subtract the daily global area-weighted mean Z500 from the daily Z500 data at each grid point so that there is no linear trend in the Z500 data due to warming (Christidis and Stott, 2015; Siler et al., 2019; Zhuang et al., 2021b). To mitigate high-frequency synoptic noise, we employ a simple 5-day moving average filter to both Z500 and IVT."

Second, in L197-199, to apply circulation analogue to each BMU node, the authors regress total precipitation anomalies onto the circulation anomalies (PC values), and multiply that regression coefficients to the circulation anomalies (PC values) to get dynamical precipitation changes. I think this can overemphasize the contribution from the circulation anomalies, and authors may need to come up with another approach for this step. The detail is written below.

In previous studies for circulation analogue, the linear coefficients or subsamples to estimate the dynamic precipitation anomalies are calculated only using the circulation anomalies, as you wrote in L205-206. However, here, total precipitation anomalies are directly regressed onto the circulation anomalies in same BMU node to get coefficients, and that regression coefficients are used to estimate the dynamic precipitation anomalies. This is like assuming that total precipitation anomalies are similar to anomalies from the moist circulation (or dynamic contribution) even before the decomposition. I think this would artificially overemphasize the dynamic contribution. So, I recommend you to revise step 4 in L197-199. You may use the linear coefficients for reconstructing target-day circulation from analogue-day circulations to getting target-day dynamic precipitation anomalies from the analogue-day precipitation anomalies in same BMU node, following Deser et al. 2016. I think you should avoid to directly link the total precipitation anomalies to the circulation anomalies to get the coefficients for estimating dynamic contribution.

Sorry for the confusion. In the original manuscript, we have a paragraph briefly describing the rationale behind regressing P' onto circulation anomalies (right after the detailed method steps). In order to help readers better understand why we calculate it this way and how it differs from previous methods (especially constructed flow analogue), we move that paragraph above the method steps and revise it to include some equations for better clarification. Related texts are attached here.

Line 201-217: "The basic idea behind SOMA is to incorporate variability in $P'_{SOMA}$ for days with the same BMU node, similar to the flow analogue method, while adhering to the C2007 framework. In contrast to the flow analogue method, where analogue days are determined by minimizing the Euclidean distance of anomalous circulation patterns, in SOMA, the analogue days are directly selected from the daily samples sharing the same BMU. Additionally, the constructed flow analogue approach relies on a linear combination of precipitation anomaly from analogue days to quantify circulation contribution at a target day $j$, i.e., $P'_{j,dyn} = \sum_i a_i P'_i = \sum_i a_i P'_{i,dyn} + \sum_i a_i P'_{i,res}$, where the coefficients $a_i$ are determined by the linear dependency of the circulation pattern on target day $j$ and those on analogue days $i$, i.e., $Z'_j = \sum_i a_i Z'_i$. However, it is important to note that the analogue day precipitation anomaly $P'_i = P'_{i,dyn} + P'_{i,res}$ includes both a circulation-contributed component and a residual component. This approach overlooks the potential thermodynamic influences in the precipitation anomaly on analogue days, and the resulting circulation-contributed component $P'_{j,dyn}$ still retains a thermodynamic residual $\sum_i a_i P'_{i,res}$, which may not be negligible with limited samples. To mitigate this issue, we make the assumption that for each group of similar circulation pattern (same SOM node), $P'_{j,dyn} = \sum_i a_i P'_{i,dyn}$. This differs from the equation form in constructed flow analogue and excludes the term $\sum_i a_i P'_{i,res}$. Following constructed flow analogue, where each circulation pattern is linearly dependent on other circulation patterns within the same SOM node, i.e., $Z'_j = \sum_i a_i Z'_i$, we can see that $P'_{dyn}$ and $Z'$ share the same linear form and coefficients. Therefore, $P'_{dyn}$ and $Z'$ can be

considered linearly dependent as well, allowing us to treat the calculation of $P'_{dyn}$ as a regression problem, which can be resolved using the steps outlined below."

PDO-related analysis

The precipitation anomalies related to PDO is often linked to the large-scale circulation over the North Pacific (retreat of Aluetian Low/ expansion of subtropical high), and it was less linked to the circulation anomalies within the US. This study can further investigate the teleconnection impact of PDO on the circulation anomalies within US, which will be the valuable contribution for regional teleconnection studies. However, the regional circulation and precipitation anomalies related to PDO will probably sensitive to the detailed pattern of tropical SST pattern (as it is known that the teleconnection within US depends on ENSO diversity), and they will be affected by the other variability (e.g., AMO, ENSO) as well. In this situation, the mere difference of positive and negative phase of PDO with relatively short length of observation would not sufficient to convince the PDO impact in this study. These limitation needs to be addressed in the manuscript.

Thank you for pointing out these limitations. And we agree that it would be impossible to disentangle the PDO contributions from various climate modes simply by using our current form of methodology. We added discussion about these limitations in the discussion section (attached below). Hope this can address your concerns.

Line 471-482: "Furthermore, the identified PDO-related SOM nodes and $P'$ is likely also modulated by tropical SST variability. This is due to the well-established understanding that ENSO and PDO can generate similar atmospheric and oceanic anomaly patterns (e.g., Hu and Huang, 2009). Additionally, other internal climate variability modes, such as the Atlantic Multidecadal Oscillation (AMO; e.g., Hu et al., 2011), North Atlantic Oscillation (NAO; e.g., Whan and Zwiers, 2017), and Interdecadal Pacific Oscillation (IPO; e.g., Dai, 2013), can also influence these patterns. The SOM circulation patterns defined by Z500' and IVT' simply represent regional-scale manifestation of larger-scale variability simultaneously influenced by multiple internal climate modes. Attempting to isolate the individual contribution of these modes using statistical methods with limited data, such as SOM or SOMA, can be a complex challenge. Therefore, our results related to PDO contribution serves as a preliminary starting point which demonstrate the combined contribution of the SOM node patterns statistically linked to the PDO. To gain a more detail quantification of the PDO's influence excluding the effects of other climate variability modes, further research integrating both observational data and climate model output is needed."

 In spite of above limitations, I think the analysis in this manuscript can be developed further to shed light on understanding the PDO teleconnection impact. The figure 10 is a good starting point where PDO-related local circulation/precipitation anomalies can be hypothesized and analyzed in detail. I suggest the authors to provide the figure for precipitation anomalies

corresponding to figure 10, showing the precipitation anomalies linked to each circulation changes between two PDO phase (for three target regions). And then, authors may select the circulation node which shows strong precipitation anomalies, linking the PDO to that circulation and precipitation anomalies. If the PDO-induced teleconnection can explain those circulation pattern with the previously known mechanisms, authors can somewhat convince that their PDO-related precipitation anomalies are indeed PDO-induced. In summary, I think the current analysis/discussion for the PDO impact is not enough, and authors need to investigate the significant circulation pattern in Fig. 10 and try to link those to the PDO-related teleconnections to ensure the PDO impact.

Thanks for the suggestion. We have modified our method of identifying PDO-related nodes to include the contrast in circulation related precipitation ($P'_{SOMA}$) in addition to node frequency (f), so as to ensure the nodes we selected has PDO impact on precipitation. Related texts are also attached below:

Line 346-361: "To further determine how much of the dynamic contribution to the multi-decadal changes of $P'$ can be linked to PDO changes, we first investigate whether the moist circulation and precipitation anomalies associated with each node may be modulated by PDO phases. We assess whether the difference in frequency-weighted $P'_{SOMA}$ between positive and negative PDO phases is statistically significant through a Monte-Carlo test. Detailed steps are as following: a) for node $k$ in month $m$, calculate node frequency for positive PDO phase (monthly PDO index > 0.5) and negative phase (monthly PDO index < -0.5) for all years, i.e., $f_{PDO+}(k,m)$ and $f_{PDO-}(k,m)$, as well as average SOMA contributed $P'$, i.e., $P'_{SOMA,PDO+}(k,m)$ and $P'_{SOMA,PDO-}(k,m)$. b) the frequency weighted $P'_{SOMA}$ difference between positive and negative PDO phases for node $k$ in month $m$ can be calculated as $\Delta_{PDO}P'_{SOMA}(k,m) = f_{PDO+}(k,m) \times P'_{SOMA,PDO+}(k,m) - f_{PDO-}(k,m) \times P'_{SOMA,PDO-}(k,m)$, and overall PDO impact for node k can be represented by summing this difference across all months, i.e., $\Delta_{PDO}P'_{SOMA}(k) = \sum_{m=1}^{12}[\Delta_{PDO}P'_{SOMA}(k,m)]$. c) randomly shuffle the sequence of years for the PDO index data and recalculate the abovementioned statistics. d) repeat step c) for many times (10000 here), if $\Delta_{PDO}P'_{SOMA}(k,m)$ is larger than 90% of all simulated values, then for node $k$ in month $m$, $P'_{SOMA}$ is significantly larger during positive PDO phase than negative phase; in contrast, if $\Delta_{PDO}P'_{SOMA}(k,m)$ is smaller than 90% of all simulated values, then $P'_{SOMA}$ is considered significantly smaller during positive PDO phase. e) similar conclusions can be drawn for node k overall by assessing $\Delta_{PDO}P'_{SOMA}(k)$ instead of $\Delta_{PDO}P'_{SOMA}(k,m)$. The resulting nodes statistically related to PDO (for each month and overall) are shown in Fig. 9."

Line 371-382: "A significant $\Delta_{PDO}P'_{SOMA}$ value could be due to differences in node frequency, node mean $P'_{SOMA}$, or both, between the two PDO phases. Our result in Fig. 9 indicates that, for most nodes, PDO phases can have very different influences on precipitation through moist circulation conditions identified in this study, for different seasons and regions. For example, node A1, characterized by geopotential high anomalies along the west and east coasts and low anomalies over the central US, mirrors a Pacific North America (PNA)-like pattern, previously

linked to Pacific variability and its impact on precipitation in the western US and GP (e.g., Ciancarelli et al. 2014; Zhuang et al. 2021a). Interestingly, node A1 does not display a significantly higher frequency during the positive PDO phase compared to the negative phase (even lower in January to April; Fig. 10). However, it stands out as one of the nodes exhibiting the most substantial contrast in $P'_{SOMA}$ between the two PDO phases over the SGP. This contrast arises because the composite circulation pattern for node A1 during the positive PDO phase exhibits stronger negative Z500' over the central US, which can promote upward motion and is generally linked to reduced convection inhibition (Fig. S5), thereby leading to larger $P'_{SOMA}$."

MINOR (TECHNICAL) COMMENTS:

L60: Is there any need to use the term "surface parameter"? I think you can just directly mention the precipitation instead of "surface parameter such as precipitation". It would be great if the terminology in this paragraph is adjusted accordingly.

We use "surface parameter" because this method and similar ones are applicable to variables other than precipitation as shown in previous studies, such as temperature and VPD. But we agree simpler terminology should be used for better flow. So in this revision, we keep "surface parameter" in Line 59 when SOM is first introduced but simply use P' afterwards for elsewhere in the manuscript. Hope this can address your concern.

L60-63: I think it is hard for readers to understand Cassano et al. 2007 (C07) SOM method by reading few sentences here. I suggest you to add and explain the equation 1 and 3 of C07, since they are quite simple. Those equations may be added on the line with simple explanations.

Thanks for the suggestion. We have revised Line 59-64:

"The idea is that the total precipitation anomaly for a certain node $K$ during a period ($P'_K$) can be decomposed as the sum of a circulation or dynamic component ($P'_{K,dyn}=\Delta f_K P'_K$) controlled by the mean frequency change of SOM nodes, a thermodynamic component ($P'_{K,the}=f_K \Delta P'_K$) related to the mean state change of precipitation for the same SOM node, and an interaction term ($P'_{K,int}= \Delta f_K \Delta P'_K$) related to both. Detailed descriptions of this method can be referred to Cassano et al. (2007) and Horton et al. (2015)."

L138: I think it is better to represent the fraction of variances only for the Z500 and IVT combined variability since the EOF analysis is done simultaneously to two variables.

Yes, we agree. PC number cutoff is based on the combined variance, we just added more information about how much variance in each variable is explained. We revise the sentence to better reflect this:

Line 144-145: "Specifically in this study, we retain the top 26 PCs which explain 87.3% of the combined variance in Z500′ and IVT′ (or 90.6% and 77.4% of the variance in Z500′ and IVT′, respectively)."

L164: "issmall" to "is small"?

Corrected.

L194-196: Why do you find the analogue day within 91 calendar windows? This might be due to the seasonal dependence of circulation-precipitation relationship, and it is better to mention this somewhere in the text.

Thanks, added in Line 226-227:

"3) For any day t0, assuming its BMU is node i, we find all days tj (j = 1, 2, …, M) that have this node i as their BMU and are within the 91-day calendar window centered on the day t0 but not in the same year as t0, to account for the seasonal dependence of circulation-precipitation relationship, i.e., BMU(xtj)=BMU(xt0), |DOY(tj) − DOY(t0)| ≤ 45 d, Year(tj)≠Year(t0)."

L317: "Fig. 9 (1st column) shows the period mean P′ for different seasons and regions" à There are no seasonal information in Fig. 9. You may need to change the text.

Thanks, "different seasons" is removed.

L323: "Table2" to "Table1"?

Corrected.

---

## Author Response (AR2)

**Reply to Reviewer 1's Comments:**

GENERAL COMMENTS:

I deeply appreciate for author's efforts. After revisions, the manuscripts become clearer and the scientific significance is much more improved. Especially, I can feel that analysis and discussion regarding the PDO impact become way better than before. While I'm now agreeing with most of author's results, there are some additional points which need to be clarified. After the minor revisions, I'm happy to do reviewing again for publication.

Issues related to detrending

Now I can clearly see that high-frequency noise is removed via the 5-day moving average filter. However, still, I don't know whether the data is detrended or not. You mentioned that Z500 is subtracted by global-mean, but I think it is extraction of pattern effect rather than the detrending. Since global warming entails some tropical SST pattern and related Z anomalies, I cannot agree that removing global-mean is same with detrending of forced response. In addition, since you include the IVT for the circulation anomalies, I think it inevitably has the effect of the global warming. Indeed, in L406-408 and L416-419, you attribute difference between past and recent PDO impact to the global warming. So, I think your manuscript itself admit that there are forced responses which are not detrended.

Since the analysis regarding the forced responses (such as L406-408 / L416-419) is very interesting, I think it would be great to keep current method. However, since it is different from previous circulation analogue studies, I suggest authors to explicitly mention that the anomalies are not detrended and include the forced responses by global warming.

Thank you very much for this comment! We further explain this in Lines 98-105 (note that all line numbers hereafter refer to the track-changes version of the revised draft):

"To account for the thermal expansion of the warming atmosphere, we subtract the daily global area-weighted mean Z500 from the daily Z500 data at each grid point so that there is no linear trend in the Z500 data due to warming (Christidis and Stott, 2015; Siler et al., 2019; Zhuang et al., 2021b); however, this procedure does not remove any anthropogenically forced changes in atmospheric circulation patterns. Other studies (e.g., Sippel et al. 2019) also use linear detrending or high-pass filtering to preprocess Z500 data at each grid point, assuming that the forced circulation response was smooth and additive; we do not employ these approaches due to our short observational record and they could remove some decadal circulation trend due to internal variability (Zhuang et al. 2021b)."

Analogue part in SOMA methodology

First, it become much easier to read and understand the methodology, which is especially important for this study with new techniques. I have some minor comment with the part 4) in page 7 (L229-231). After getting analogue days from each BMU (SOM) nodes, the authors get regression coefficients by regressing precipitation anomalies directly onto the circulation anomalies. This is definitely different with previous circulation analogue such as Lehner et al. 2018, which get the linear coefficients by finding optimal combinations of analogue-day circulation anomalies to reconstruct target day circulation anomalies. The method in this study (regressing precipitation directly onto the circulation anomalies) would inherently maximize the contribution from the moist circulation, and I thought this would be problematic. However, the high R2 values in Figs. 5,6 would be only possible if the circulation anomalies are indeed important as their assumption, so I'm now agreeing with your regression methods. However, I think you need to

explicitly mention that "solving regression problem for P" in this study is different from the circulation analogue in previous studies (Deser et al. 2016; Lehner et al. 2018) for clarity.

Thank you for the suggestion. We added this statement to Lines 223-226:

"Therefore, $P'_{SOMA}$ and $Z'$ can be considered linearly dependent as well, allowing us to treat the calculation of $P'_{SOMA}$ as a regression problem, which is different from the circulation analogue in previous studies (e.g., Deser et al. 2016; Lehner et al. 2018). Detailed steps are outlined below."

Other minor comments:

L11-12: I think the content of "and 62~68% of the amplitude of the mean precipitation anomalies" are quite redundant for abstract. I think this is estimation from beta in Fig. 5b, and similar to results in Fig. 5a, which is already mentioned as 54-61%. I suggest the removal of this phrase.

Done!

L18-19: It seems the conjunction is not appropriate here. If I were you, I will use

"However, these circulation-induced changes are not totally related to the PDO phase shift (mostly less than half) since internal variability or anthropogenically induced changes in circulation can also be potential contributors."

Done!

L35-37: After revision, you decided to use "moist circulation vs. residual" rather than "dynamic vs. thermodynamic". But here, still, you have dynamic vs. thermodynamic perspective. It would be better to change the terms "dynamic vs. thermodynamic" to "moist circulation vs. residual" in all possible places in manuscript for the consistency.

Thank you for the comment. Here, we deleted "(dynamics)" and "(thermodynamics)" as you suggested (Lines 36-37). However, we keep the use of "dynamics" and "thermodynamics" in the literature review section (Lines 52-78) as these are the terms used in the articles referenced.

Following that, we explain the use of "moist circulation vs. residual" instead of "dynamic vs. thermodynamic" in Lines 200-207: "It's worth noting that attempting to separate the actual dynamic and thermodynamic components solely through a circulation clustering approach like SOM can be challenging. Each type of circulation, as represented by SOM nodes, inherently encompasses thermodynamic responses. Therefore, the distinction between "dynamic" and "thermodynamic" components can be ambiguous when using these terms. Consequently, we prefer to refer to the precipitation anomaly influenced by moist circulation patterns involving Z500 and IVT as $P'_{SOMA}$ instead of $P'_{dyn}$, and the residual part that cannot be explained by circulation as $P'_{RES}$ instead of $P'_{the}$. This emphasizes that our results regarding circulation contributions are contingent on our chosen set of circulation variables.".

L38-39: I think this study does not attribute the recent trend to anthropogenic warming. This sentence is now adding the confusion.

By assuming that circulation changes are totally due to natural variability, we can infer the anthropogenic warming's contribution to the recent trend (or its lower bound, as in Zhuang et al. 2021b). However, since this is not the focus of this study, and we agree these statements could add more confusion, we have deleted Lines 38-44 in the introduction.

L49-51: I suggest "atmospheric circulation variability or the PDO-related circulation variability" -> "the PDO-related circulation variability". Atmospheric circulation variability itself contains the effect from PDO-related circulation anomalies, and this paragraph seems to be more related to the PDO.

Revised.

L70-73: In Lehner et al. 2018, they use monthly data for circulation analogue, rather than daily data. You need other references which use daily data for circulation analogue. If there are no such study, you may need to mention the time-scale difference.

I've added a few more references that use constructed circulation analogue at daily scale at Lines 69-70:

"Flow analogue or dynamic adjustment (e.g., Deser et al., 2016; Horowitz et al., 2022; Jezequel et al., 2018; Lehner et al., 2018; Terray, 2021; Yiou et al., 2007) …".

L80: "daily variability and amplitude of anomaly" -> "daily variability", since they are similar.

Revised.

L85: "precipitation anomalies, their variability" -> "daily variability". In abstract or introduction, giving both of them is adding confusions.

Revised.

L185-187: Could you explain why the fact that "flow analogue explains a larger portion of observed P'" is the reason for "the flow analogue is better estimate of circulation contribution compared to the C2007"? To me, "larger portion of observed P'' just means that circulation analogue has larger contribution from the circulation component. I agree with that flow analogue has advantage for the flexible precipitation anomalies to each circulation type, but it is hard to agree with this particular part.

We revised Lines 189-193:

"The constructed flow analogue method generally explains a larger portion of the observed $P'$ variance and its estimate is not sensitive to SOM node number, so it has a higher accuracy in capturing the complex relationship between atmospheric circulation and precipitation anomalies and theoretically provides a better estimate of circulation contribution compared to the C2007 method."

L206-L217: In previous paragraph, author said that they will use the term P_SOMA instead of P_dyn. However, in this paragraph, authors are using P_dyn. It is now adding confusions.

We used "$P'_{dyn}$" in this paragraph because we were referring to the analogue precipitation anomaly calculated by the traditional constructed analogue approach, instead of SOMA. This time we have changed them all to $P'_{SOMA}$ to avoid confusion in the revised manuscript.

L222: The subscript "i" is used for analogue day in L207 but suddenly used for BMU nodes. It would be better to use different one.

We have revised the use of subscripts in Lines 227-244 so now "i" represents days and "n" represents nodes.

---

## Author Response (AR3)

**Public justification (visible to the public if the article is accepted and published)**:
Dear authors,

Thank you for addressing the remaining comments by the reviewer. In terms of the text and figures, the manuscript is now ready for publication. However, I noticed that the article is missing a statement on Data availability. Please review ACP's data policy (https://www.atmospheric-chemistry-and-physics.net/policies/data_policy.html) and add this statement.

Best regards
Corinna Hoose

Dear Dr. Hoose,

We greatly appreciate your feedback and are excited to know that our manuscript is nearing readiness for publication.

Following your instruction, we have added the "Data availability" section. We have also added the "Code availability" and "Author contribution" as instructed in the submission system. Additionally, we have acknowledged an extra funding source that supported our research. We have added a few minor grammatical edits. All these changes are marked in the track-changes version of the revised manuscript.

Thank you again for your guidance throughout this process and we look forward to hearing back from you soon.

Best,

Yizhou Zhuang & Rong Fu